# Paracrine FGFs target skeletal muscle to exert potent anti-hyperglycemic effects

Lei Ying[1,2,8], Luyao Wang[1,8], Kaiwen Guo[1,8], Yushu Hou[1,8], Na Li[1,2], Shuyi Wang[3], Xingfeng Liu[4,5,6], Qijin Zhao[4,5,6], Jie Zhou[1], Longwei Zhao[1], Jianlou Niu[1], Chuchu Chen[1], Lintao Song[1], Shaocong Hou[4,5,6], Lijuan Kong[4,5,6], Xiaokun Li[1], Jun Ren [3✉], Pingping Li [4,5,6✉], Moosa Mohammadi [7✉] & Zhifeng Huang [1✉]

Several members of the FGF family have been identified as potential regulators of glucose homeostasis. We previously reported that a low threshold of FGF-induced FGF receptor 1c (FGFR1c) dimerization and activity is sufficient to evoke a glucose lowering activity. We therefore reasoned that ligand identity may not matter, and that besides paracrine FGF1 and endocrine FGF21, other cognate paracrine FGFs of FGFR1c might possess such activity. Indeed, via a side-by-side testing of multiple cognate FGFs of FGFR1c in diabetic mice we identified the paracrine FGF4 as a potent anti-hyperglycemic FGF. Importantly, we found that like FGF1, the paracrine FGF4 is also more efficacious than endocrine FGF21 in lowering blood glucose. We show that paracrine FGF4 and FGF1 exert their superior glycemic control by targeting skeletal muscle, which expresses copious FGFR1c but lacks β-klotho (KLB), an obligatory FGF21 co-receptor. Mechanistically, both FGF4 and FGF1 upregulate GLUT4 cell surface abundance in skeletal muscle in an AMPKα-dependent but insulin-independent manner. Chronic treatment with rFGF4 improves insulin resistance and suppresses adipose macrophage infiltration and inflammation. Notably, unlike FGF1 (a pan-FGFR ligand), FGF4, which has more restricted FGFR1c binding specificity, has no apparent effect on food intake. The potent anti-hyperglycemic and anti-inflammatory properties of FGF4 testify to its promising potential for use in the treatment of T2D and related metabolic disorders.

[1] School of Pharmaceutical Sciences, Wenzhou Medical University, Wenzhou, Zhejiang 325035, China. [2] School of Basic Medical Sciences, Wenzhou Medical University, Wenzhou, Zhejiang 325035, China. [3] Department of Cardiology and Shanghai Institute of Cardiovascular Diseases, Zhongshan Hospital, Fudan University, Shanghai 200032, China. [4] State Key Laboratory of Bioactive Substance and Function of Natural Medicines, Institute of Materia Medica, Chinese Academy of Medical Sciences and Peking Union Medical College, Beijing 100050, China. [5] Diabetes Research Center, Chinese Academy of Medical Sciences and Peking Union Medical College, Beijing 100050, China. [6] CAMS Key Laboratory of Molecular Mechanism and Target Discovery of Metabolic Disorder and Tumorigenesis, Chinese Academy of Medical Sciences and Peking Union Medical College, Beijing 100050, China. [7] Department of Biochemistry & Molecular Pharmacology, New York University School of Medicine, New York, NY 10016, USA. [8] These authors contributed equally: Lei Ying, Luyao Wang, Kaiwen Guo, Yushu Hou. ✉email: jren_aldh2@outlook.com; lipp@imm.ac.cn; mohammadimoosa@gmail.com; hzf@wmu.edu.cn

The mammalian fibroblast growth factor (FGF) family comprises 18 structurally-related polypeptides (FGF1-FGF10 and FGF16-FGF23); these act in a paracrine or endocrine manner to mediate a plethora of critical events in development, tissue homeostasis/repair, and metabolism[1,2]. The recent discovery of essential roles for FGFs in the regulation of homeostasis of glucose, lipid, cholesterol, and adipose tissue remodeling has brought FGF signaling to the forefront in terms of drug discovery for type 2 diabetes mellitus (T2D) and related metabolic diseases[3–5]. To date, two endocrine FGFs (i.e., FGF21 and FGF19) and a single paracrine FGF (FGF1) have been shown to exert multiple beneficial effects in obesity and T2D, including potent glucose-lowering activity, improved lipid profiles, and enhanced energy expenditure[3–5]. Notably, these FGFs mitigate a number of complications in T2D[4,6] without eliciting many of the side effects associated with current T2D medications; for example, thiazolidinediones (TZDs) tend to engender weight gain, bone loss, and edema[7].

FGFs implement their diverse actions by binding, dimerizing, and consequently activating their cognate FGF receptors (FGFRs) in an heparan sulfate (HS)-dependent (in the case of paracrine FGFs)[2,8] or an HS and Klotho co-receptor dependent (in the case of endocrine FGFs) manner[2,9,10]. In mammals, there are four distinct FGFR genes (FGFR1-4)[2]. Tissue-dependent alternative splicing of FGFR1-3 genes gives rise to epithelial "b" and mesenchymal "c" isoforms with distinct FGF binding specificity thus elaborating the number of principal FGFRs to seven[11]. Upon activation, FGFR triggers activation of multiple downstream signaling pathways, including PLCγ/PKC, FRS2α/RAS-MAPK, FRS2α/Gab1/PI3 kinase/Akt, CrkL/Cdc42-Rac and CaMKK2/AMPKα and LKB1/AMPKα[12–15], thereby reprogramming the cellular transcriptional landscape that ultimately determines cellular fate/response.

We recently showed that a low threshold of FGF-induced FGF receptor (i.e., FGFR1c) dimerization is sufficient to elicit glucose-lowering activity[16,17]. We therefore surmised that ligand identity may not matter, and that other cognate paracrine FGFs of FGFR1c might also possess anti-hyperglycemic activity in vivo. If so, these could be repurposed for treating T2D and related metabolic disorders. Indeed, by testing the glucose-lowering activity of representative members from the five paracrine FGF subfamilies in T2D mice, we identified FGF4 as a new anti-hyperglycemic molecule. We show that both paracrine FGF4 and FGF1 elicit more potent and durable anti-hyperglycemic effects than endocrine FGF21. We attribute the superior glycemic control of paracrine FGFs to the ability of these FGFs to stimulate glucose disposal in skeletal muscle, a β-klotho (KLB)-deficient tissue that is off-limits for endocrine FGF21. We further show that paracrine FGFs exert their glycemic control by activating the AMPK signaling pathway in an insulin-independent manner.

## Results

**Multiple cognate FGFR1c ligands can normalize glucose in T2D mice.** In addition to the FGF1 subfamily, the paracrine FGF4, FGF9, and FGF8 subfamilies are cognate ligands of FGFR1c[18,19]. Accordingly, we expressed and purified three human recombinant FGFs [FGF4 (rFGF4), FGF8 (rFGF8), and FGF9 (rFGF9)]—all founding members of their respective subfamilies (Fig. 1a)—and tested their anti-hyperglycemic activities following intraperitoneal (i.p.) injection into db/db mice. Recombinant FGF1 (rFGF1) (the founding member of the FGF1 subfamily) and FGF7 (rFGF7) (the founding member of the FGF7 subfamily that does not recognize FGFR1c) were used as positive and negative controls, respectively. Reminiscent of rFGF1—an established anti-hyperglycemic ligand[5]—rFGF4,

rFGF8, and rFGF9 all elicited a glucose-lowering effect (Fig. 1b), with rFGF4 exerting the greatest response. In contrast, rFGF7 was devoid of any such property (Fig. 1b).

We next compared the anti-hyperglycemic effects of paracrine rFGF4 and rFGF1 and endocrine rFGF21 (all at 1.0 mg/kg body weight) in db/db, DIO, and ob/ob mice over the course of 36 h. Both rFGF4 and rFGF1 elicited significant glucose-lowering effects 3 h after administration and normalized the glucose level by 6 h in these mice (Fig. 1c–e). These anti-hyperglycemic effects persisted for at least 24 h. However, the corresponding effect was much weaker in both amplitude and duration in rFGF21-treated mice, tapering off after 6 h (Fig. 1c–e). Notably, as previously reported for rFGF21[20,21], rFGF4 had no impact on food intake (Fig. 1f–h) whereas rFGF1 did inhibit food intake consistent with previously reported data[5] (Fig. 1f–h). We also found that administration of either rFGF1 or rFGF4 potently remedied high glucose in fasted db/db mice (Supplementary Fig. 1a). As in the case of FGF1[5], a single dose of rFGF4 rescued hyperinsulinemia in db/db mice (Supplementary Fig. 1b) but had no significant effect on either blood glucose or insulin levels in chow-fed wild type mice (Supplementary Fig. 1c, d), indicating that exogenous FGF4 does not stimulate pancreatic β-cell insulin release. Moreover, a bolus of rFGF4 as high as 3 mg/kg body weight did not induce hypoglycemia in db/db mice (Supplementary Fig. 1e). Examination of the pharmacokinetics of rFGF4 following a single injection into Sprague Dawley (SD) rats showed the in vivo half-life to be ~4.4 h, which was approximately 1.8 times greater than that of rFGF1 (~2.5 h) (Supplementary Table 1).

**Acute rFGF4 treatment upregulates GLUT4 expression and translocation in skeletal muscle.** To understand the underlying mechanism(s) for the dramatic differences in glucose-lowering potencies between paracrine FGFs and endocrine FGF21, we studied their target organ(s) via micro positron emission computed tomography (microPET-CT). To do so, db/db mice were treated acutely (i.e., single bolus) with rFGF4, rFGF1, rFGF21, or buffer (as a control) for 2 h, then infused with radiolabeled fluodeoxyglucose ($^{18}$F-FDG) and its uptake monitored for 1 hr by whole body positron emission tomography (PET). We found a pronounced enrichment of $^{18}$F-FDG in skeletal muscle (including in both the back (erector spinae muscles) and lower limb muscles (thigh muscles)) of rFGF4 and rFGF1 treated animals (Fig. 2a–c). In contrast, rFGF21 did not cause any such enrichment in muscle tissues (Fig. 2a–c). These results were corroborated upon measuring the radioactivity of excised organs via gamma-counting: $^{18}$F-FDG uptake in skeletal muscle was robust and comparable in both rFGF4 and rFGF1-treated db/db mice, whereas rFGF21-treated animals showed no such enrichment (Fig. 2d).

The glucose transporter GLUT4 is the principal mediator of glucose uptake in skeletal muscle[22,23]. We therefore examined the effect of single-dose FGF administration on the expression and cell surface translocation of GLUT4 in db/db mice. Indeed, we found that administration of rFGF4 or rFGF1 induced a profound upregulation of GLUT4 transcription 2 h post administration (Supplementary Fig. 2a), which manifested in an enhancement in total GLUT4 expression that persisted for at least 6 h after administration (Fig. 2e, g and Supplementary Fig. 2b). These effects were accompanied by a conspicuous translocation of GLUT4 to the skeletal muscle cell membrane (Fig. 2e, f). Notably, administration of the GLUT inhibitor phloretin[22] abolished rFGF4's acute glucose-lowering effects in db/db mice (Supplementary Fig. 2c). In contrast to rFGF4 and rFGF1, rFGF21 failed to upregulate GLUT4 expression/translocation in skeletal muscle (Fig. 2e–g). These results are consistent with the fact that this tissue lacks expression of KLB (Fig. 2h), an obligatory co-receptor

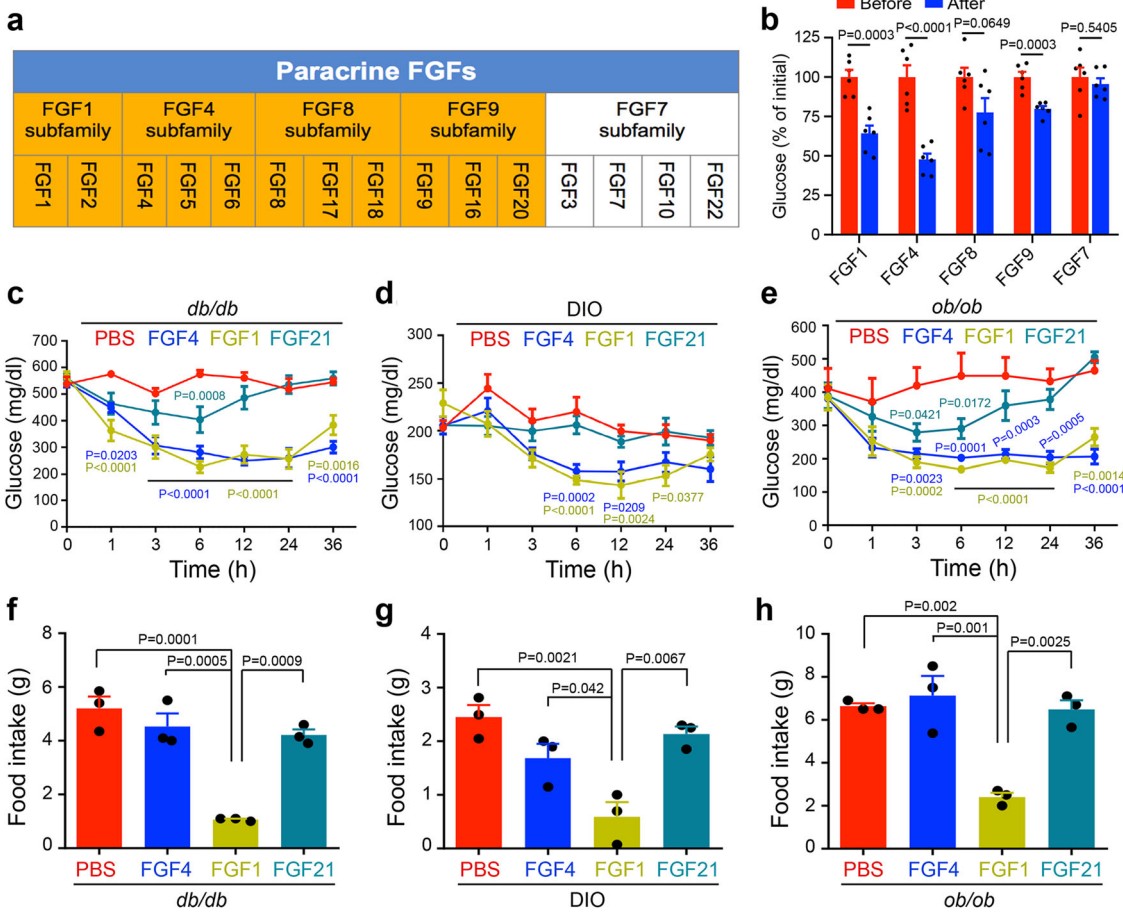

**Fig. 1 Multiple cognate FGFs of FGFR1c are capable of lowering blood glucose levels in diabetic mice. a** Schematic of paracrine FGF subfamilies; those that can bind FGFR1c are highlighted in yellow. **b** Changes in blood glucose levels in *db/db* mice 6 h after a single intraperitoneal (i.p.) injection of rFGF1, rFGF4, rFGF8, rFGF9, or rFGF7 (each at 1.0 mg/kg body weight) (n = 6). **c–e** Blood glucose levels in *db/db* (**c**) (n = 6), DIO (**d**) (PBS (n = 6), rFGF4 (n = 7), rFGF1 (n = 5), rFGF21 (n = 6)), and *ob/ob* (**e**) (PBS (n = 5), rFGF4 (n = 4), rFGF1 (n = 5), rFGF21 (n = 5)) mice at various times after acute i.p. injection of rFGF4, rFGF1, rFGF21 (each at 1.0 mg/kg body weight) or PBS buffer control. The statistical comparisons in these figures are PBS vs FGF4, PBS vs FGF1, and PBS vs FGF21, respectively. **f–h** Food intake of *db/db* (**f**), DIO (**g**), and *ob/ob* (**h**) mice measured 24 h after acute i.p. injection with rFGF4, rFGF1, rFGF21 (each at 1.0 mg/kg body weight), or with PBS as a buffer control (n = 3). Data are presented as mean +/− SEM. Statistical comparison in (**b**) is unpaired two-tailed test. Statistical comparisons in (**c–e**) are two-way ANOVA tests with Tukey's multiple comparisons tests. Statistical comparisons in (**f–h**) are one-way ANOVA tests with Tukey's multiple comparisons tests. Source data are provided as a Source data file.

for FGF21[10]. We conclude that paracrine FGF1 and FGF4 mediate their potent anti-hyperglycemic effects by targeting KLB-deficient skeletal muscle.

**rFGF4 induces GLUT4 expression and translocation by activating the AMPKα pathway.** GLUT4 expression and translocation in skeletal muscle is regulated by activation of AMPKα or by insulin signaling[24,25]. We detected no significant changes in insulin signaling in the skeletal muscle of *db/db* mice 6 h post-administration of a single dose of rFGF4 or rFGF1 as measured by phosphorylation of IRS1 and/or AKT—two known reporters for insulin signaling (Supplementary Fig. 3a, b). Surprisingly, both rFGF4 and rFGF1 induced robust activation of AMPKα in *db/db* mouse skeletal muscle (Fig. 3a and Supplementary Fig. 3c), implying that these ligands stimulate GLUT4 expression and translocation by activating AMPKα independently of insulin signaling. To test this conjecture, we pretreated *db/db* mice with the selective AMPKα antagonist compound C[26] prior to administration of rFGF4 or rFGF1. This pretreatment diminished the ability of rFGF4 or rFGF1 to induce GLUT4 expression and translocation (Supplementary Fig. 4a, b and Supplementary Fig. 4d, e) as well as anti-hyperglycemic activity (Supplementary Fig. 4c and Supplementary Fig. 4f). These data were

further corroborated using mice lacking AMPKα2, the principal AMPK isoform in skeletal muscle[24]. In these animals, the acute glucose-lowering effect of rFGF4 or rFGF1 was dramatically compromised (Fig. 3b and Supplementary Fig. 5a), and was accompanied by a marked reduction in GLUT4 expression and translocation (Fig. 3c, d and Supplementary Fig. 5b, c).

FGFR1c is the most abundantly expressed FGFR isoform in skeletal muscle (Fig. 3e)[19,27], implying that rFGF4 and rFGF1 upregulate GLUT4 expression/translocation and induce anti-hyperglycemic activity by binding and activating FGFR1c. Indeed, rFGF4 incurred a major loss in its ability to upregulate AMPKα activation and GLUT4 expression/translocation when administered to mice containing a muscle-specific FGFR1 deletion (Fig. 3f, g). Accordingly, rFGF4 failed to correct hyperglycemia in these mice (Fig. 3h). Consistent with these data, the FGFR1 kinase antagonist PD166866[28] blocked rFGF4's acute glucose-lowering effect (Supplementary Fig. 5d). Importantly, PD166866 also abolished the anti-hyperglycemic activity of rFGF1 implying that rFGF1 also activates FGFR1c to stimulate GLUT4 translocation and AMPKα activation (Supplementary Fig. 5e-g).

Consistent with the observed insulin-independent effects of rFGF4 on skeletal muscle in vivo, we found that rFGF4 is

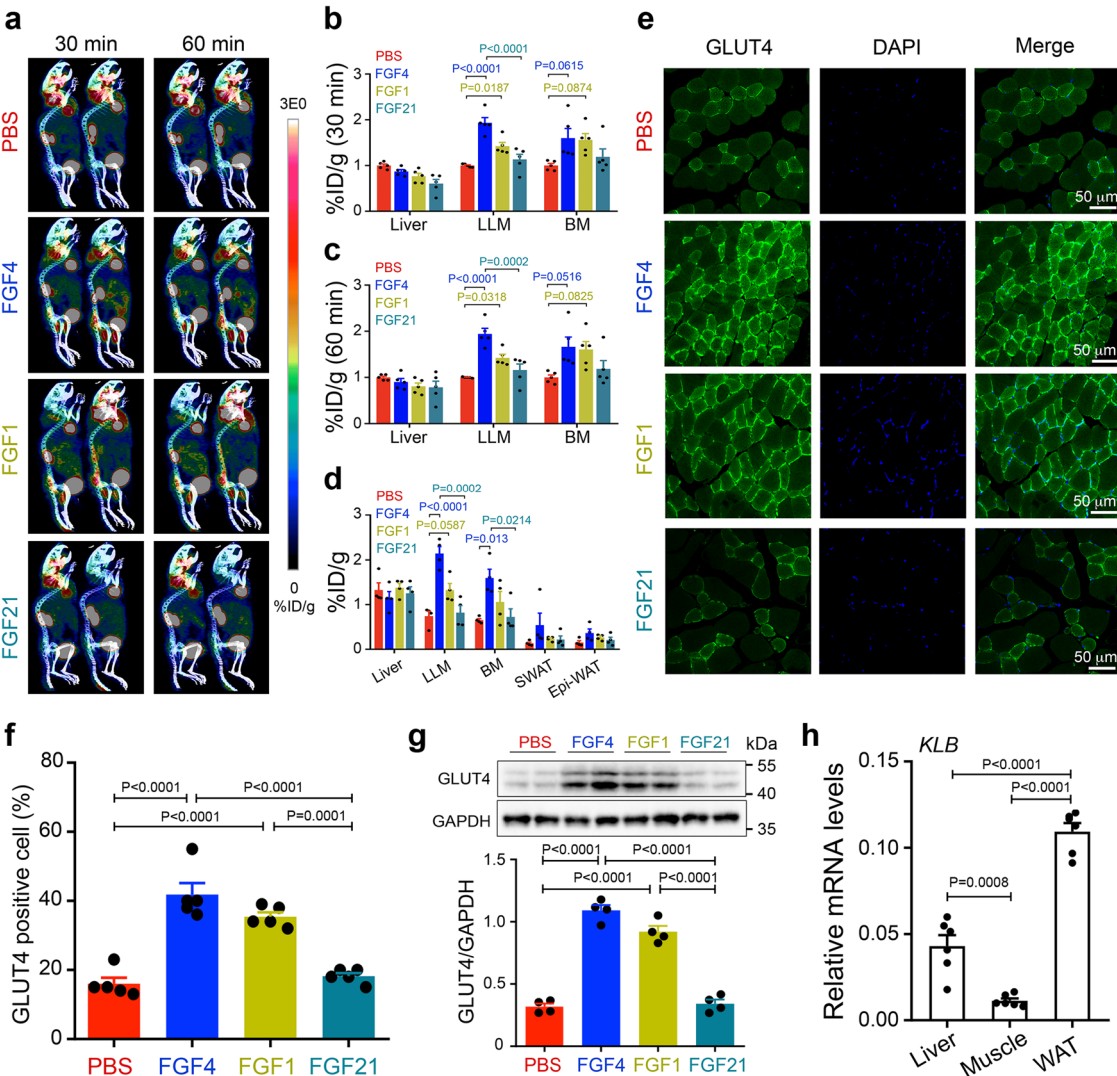

**Fig. 2 Acute glucose-lowering effect of rFGF4 and rFGF1 induces GLUT4 protein expression and translocation in skeletal muscle. a–c** PET imaging (**a**) and uptake of [18]F-FDG into *db/db* mouse organs 30 min (**b**) and 60 min (**c**) after infusion with radiolabeled fluodeoxyglucose ([18]F-FDG) through the lateral tail vein. All these animals received i.p. injection of rFGF4, rFGF1, rFGF21 (each at 1.0 mg/kg body weight), or buffer controls 2 h prior to infusion with [18]F-FDG ($n = 5$). Images in (**a**) are lateral views; red, high uptake; blue, low uptake. BM: back muscle; LLM: lower limb muscle. **d** [18]FDG uptake by liver, LLM, BM, subcutaneous white adipose tissue (SWAT) and epididymal white adipose tissue (Epi-WAT) of *db/db* mice 3 h after i.p. injection of buffer (control), rFGF4, rFGF1, or rFGF21 (each at 1.0 mg/kg body weight) as measured in excised tissues by gamma-ray emission ($n = 4$). **e, f** Translocation of GLUT4 in LLM of *db/db* mice 6 h after i.p. injection of PBS buffer (control), rFGF4, rFGF1, or rFGF21 (each at 1.0 mg/kg body weight) shown by immunofluorescence staining with an antibody to GLUT4 (**e**) ($n = 5$). Scale bars, 50 μm. Data were quantified using ImageJ software (**f**). **g** Expression of GLUT4 in LLM of *db/db* mice 6 h after i.p. injection of PBS buffer (control), rFGF4, rFGF1, or rFGF21 (each at 1.0 mg/kg body weight) measured by western blot analysis using an anti-GLUT4 antibody and an antibody to GAPDH as a loading control (upper panel) ($n = 4$). Data were quantified using ImageJ software (lower panel). **h** mRNA level of β-klotho (KLB) in liver, skeletal muscle and epididymal WAT (WAT) of WT mice determined by real-time PCR analysis ($n = 6$). Data are presented as mean $+/-$ SEM. Statistical comparisons in (**b–d**) and (**f–h**) are one-way ANOVA test with Tukey's multiple comparisons tests. Source data are provided as a Source data file.

sufficient to induce glucose uptake in differentiated rat myoblasts cells (L6) (Fig. 4a). In addition, dose-dependent rFGF4 treatment led to the phosphorylation and activation of FGFR1 and AMPKα, as well as an upregulation of GLUT4 expression in these cells (Fig. 4b). Indeed, rFGF4 noticeably increased the surface abundance of GLUT4 (Fig. 4c). This observation was also replicated in the context of 3T3-L1 adipocytes transfected with Myc-GLUT4-GFP expression plasmid (Supplementary Fig. 6a, b) (functioning as a model cell), in which rFGF4 robustly promoted the translocation of GLUT4 to cell membrane (Fig. 4d). Moreover, either FGFR1 inhibition by PD166866 or AMPKα2 gene silencing by siRNA significantly blocked these rFGF4-induced

activities (Fig. 4e, f). Finally, incubation with rFGF4 did not enhance the ability of insulin to induce AKT phosphorylation in either normal or insulin-resistant rat L6 myocytes (Fig. 4g, h), while exposure of L6 myocytes to insulin did enhance GLUT4 expression and translocation in an FGFR1/AMPK signaling-independent manner (Fig. 4b–d). This presumably reflects activation of the classical IRS1/AKT/GLUT4 pathway[29]. Taken as a whole, the data from these experiments lead us to conclude that a single dose of rFGF4 upregulates GLUT4 expression and translocation by activating AMPKα signaling operating downstream of FGFR. Notably, as consistent with its neutral effect on blood glucose in wild-type mice, rFGF4 did not activate AMPK or

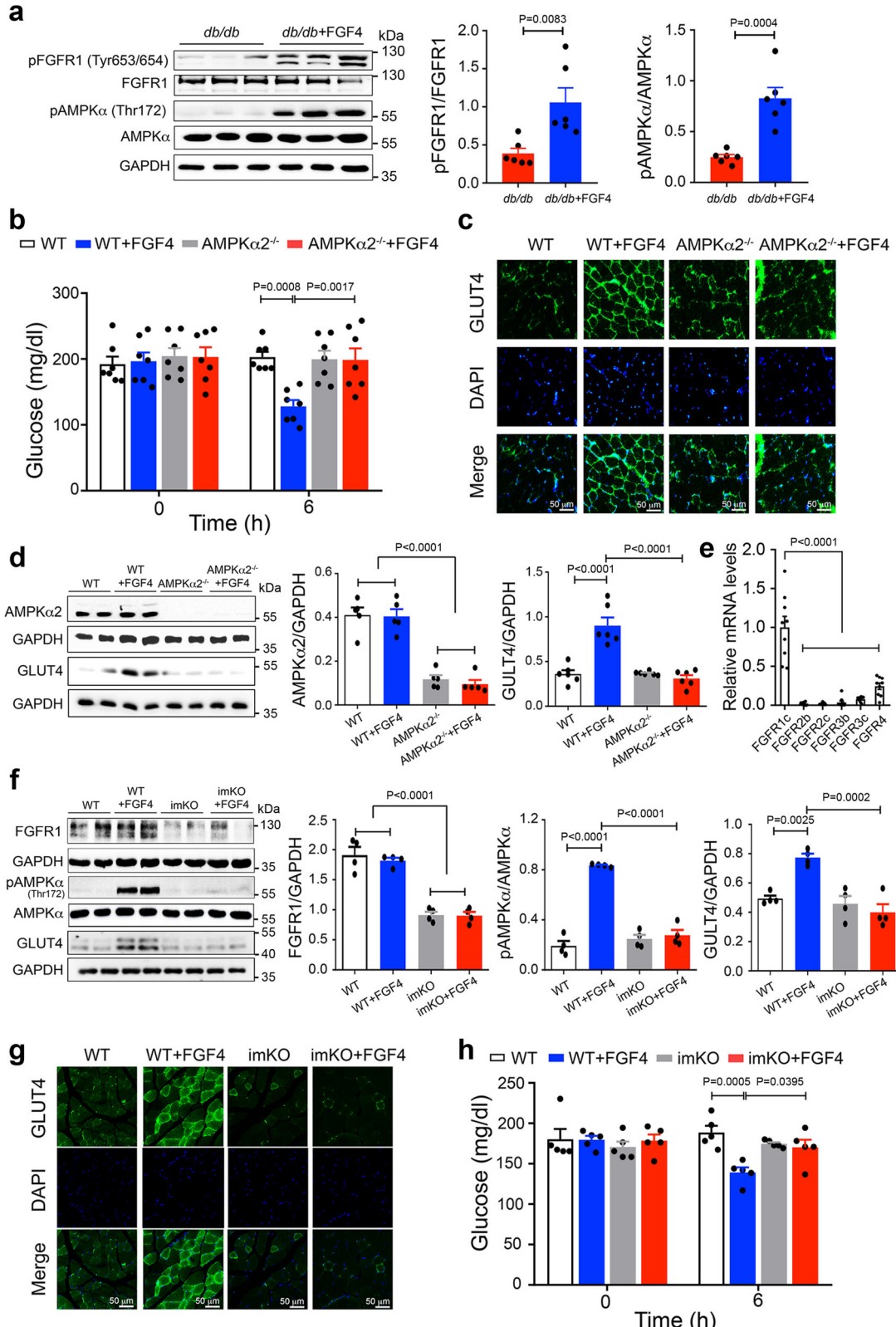

induce any change in the expression and translocation of GLUT4 in skeletal muscle in these animals (Supplementary Fig. 1f).

**rFGF4 activates AMPKα by triggering Ca²⁺/CaMKK2 signaling downstream of FGFR1.** PLCγ acts downstream of FGFR1 to hydrolyze phosphatidylinositol 4,5-bisphosphate, thereby generating

diacyl glycerol and inositol 1, 4, 5-triphosphate (IP3) second messengers[30,31]. The latter induces the release of $Ca^{2+}$ from intracellular stores[31,32]. The increase in intracellular $[Ca^{2+}]$ could lead to activation of $Ca^{2+}$/calmodulin-dependent protein kinase kinase 2 (CaMKK2, also known as CaMKKβ), and consequently of AMPKα[33,34]. We therefore surmised that rFGF4-FGFR1 signaling might activate AMPKα via the $Ca^{2+}$/CaMKK2 pathway. Indeed, we

**Fig. 3 rFGF4 induces GLUT4 expression and translocation in skeletal muscle via FGFR1/AMPKα signaling. a** Phosphorylation levels of FGFR1 and AMPKα hosphorylat*db/db* mice 6 h after i.p. injection of buffer (control) or rFGF4 (1.0 mg/kg body weight) as determined by western blotting (left panel) and quantitated using ImageJ software (center and right panels) ($n = 6$). **b** Blood glucose levels 6 h after i.p. injection of rFGF4 (1.0 mg/kg body weight) into WT and AMPKα2 knockout mice (AMPKa2$^{-/-}$) ($n = 7$). All mice were fed with a high-fat diet for 12 weeks. **c** Expression and translocation of GLUT4 in LLM of AMPKα Expressiut mice 6 h after i.p. injection of rFGF4 (1 mg/kg body weight) shown by immunofluorescence staining using an anti-GLUT4 antibody. Data are representative of 6 mice from each group. Scale bars, 50 μca. **d** Expression of AMPKαE ($n = 5$) and GLUT4 ($n = 6$) in LLM of AMPKα in LLM of AMPKAMP h after i.p. injection of buffer (control) or rFGF4 (1.0 mg/kg body weight) as measured by western blotting (left panel) and quantitated using ImageJ software (center and right panels). **e** mRNA level of FGFR1c ($n = 9$), FGFR2b ($n = 7$), FGFR2c ($n = 9$), FGFR3b ($n = 9$), FGFR3c ($n = 9$), and FGFR4 ($n = 9$) in skeletal muscle of WT mice determined by real-time PCR analysis. **f** FGFR1 and GLUT4 expression and AMPKαFGFR1 and GLUT4 expression and AMPinducible skeletal muscle FGFR1 knockout (imKO) mice 6 h after i.p. injection of buffer (control) or rFGF4 (1.0 mg/kg body weight) as determined by western blotting (left panel) and quantitated using ImageJ software (right panels) ($n = 4$). **g** Expression and translocation of GLUT4 in LLM of imKO mice 6 h after i.p. injection of rFGF4 (1 mg/kg body weight) shown by immunofluorescence staining with an anti-GLUT4 antibody. Data are representative of 5 mice from each group. Scale bars, 50 μca. **h** Blood glucose levels 6 h after i.p. injection of rFGF4 (1.0 mg/kg body weight) into imKO mice, with FGFR1$^{flox/flox}$ mice (WT) as controls ($n = 5$). All mice were fed with a high-fat diet for 12 weeks. Data are presented as mean $+/-$ SEM. Statistical comparison in (**a**) is unpaired two-tailed test. Statistical comparisons in (**b**) and (**h**) are two-way ANOVA tests with Tukey's multiple comparisons tests. Statistical comparisons in (**d**–**f**) are one-way ANOVA tests with Tukey's multiple comparisons tests. Source data are provided as a Source data file.

found that rFGF4 induced a marked upregulation of intracellular [Ca$^{2+}$] in mouse embryonic fibroblasts after 10 min treatment (Supplementary Fig. 7a, b). Consistent with these data, the acute treatment of *db/db* mice with rFGF4 robustly enhanced the activity of calmodulin-dependent protein kinase I (CaMKI) (a known reporter for CaMKK2 activity[35]) in skeletal muscle (Supplementary Fig. 7c). This activity was mostly abrogated by muscle-specific deletion of FGFR1 (Supplementary Fig. 7d). These results were further validated in L6 myocyte-based experiments in which treatment with rFGF4 led to significant activation of CaMKI, an effect that was abrogated in the presence of PD166866 (Supplementary Fig. 7e). Moreover, the selective pharmacological inhibitor of CaMKK2 (STO-609)[36] significantly blocked rFGF4-induced activation of CaMKK2 and AMPKα (Supplementary Fig. 7f). These cell-based data were further corroborated using CaMKK2 knockout (CaMKK2$^{-/-}$) mice. Specifically, rFGF4 failed to reduce hyperglycemia in high fat diet fed CaMKK2$^{-/-}$ mice (Supplementary Fig. 7g). We conclude that rFGF4-induced AMPKα activation is relayed by Ca$^{2+}$/CaMKK2 signaling acting downstream of FGFR1.

**Chronic rFGF4 treatment ameliorates insulin resistance in *db/db* and DIO mice.** We next explored the effects of prolonged (i.e., chronic) administration of FGF4 by i.p. injection of *db/db* mice with rFGF4 (1 mg/kg body weight) on alternate days over the course of 37 days. Consistent with the results of our single-dose treatment experiments, this chronic treatment elicited a sustained glucose-lowering effect without causing any significant change in body weight or undesired induction of hypoglycemia (Fig. 5a, b). Similar results were observed in DIO mice receiving 15 days of rFGF4 (1 mg/kg body weight) treatment (Fig. 5c, d). Notably, rFGF4 administration did not affect energy expenditure as measured by rates of oxygen consumption (VO$_2$) or carbon dioxide production (VCO$_2$), heat production, food intake, or in total or ambulatory activity (Supplementary Fig. 8a-g). Epididymal white adipose tissue (Epi-WAT) mass, adipocyte size, lean mass, fat mass, and organ mass were not impacted (Supplementary Fig. 9a-d and Supplementary Table 2). Additionally, chronic rFGF4 administration had no deleterious impact on either bone mineral density (BMD) or bone mineral content (BMC) (Supplementary Fig. 9e, f). Finally, *db/db* mice receiving long-term rFGF4 treatment showed no overt signs of hepatic hyperplasia as revealed by intensities of markers including proliferating cell nuclear antigen (PCNA) and Ki67 (Supplementary Fig. 10a, b).

The diminished mitogenic character of rFGF4 observed in vivo was rather expected, as rFGF4 carries an engineered N-terminal truncation (residues Ala67 to Leu206) (Supplementary Fig. 10c-e) that we intentionally introduced in order to induce decrease in the mitogenic activity predicted to be associated with the wild-type parent molecule (i.e., mFGF4) (Supplementary Fig. 10f). Specifically, the truncation in rFGF4 removes several N-terminal residues in mFGF4 necessary for stable FGFR binding and dimerization. As a result, the stability/longevity of rFGF4 induced FGFR dimer is below the threshold necessary for eliciting a mitogenic response but still sufficient for a metabolic response[17]. Indeed, surface plasmon resonance (SPR) spectroscopy experiments showed that compared to mFGF4, the binding affinity of the N-terminally truncated rFGF4 to FGFR1c isoform is weaker (Supplementary Fig. 10g, h). Accordingly, proximity ligation assay (PLA) experiments showed that relative to its wild-type parent molecule, truncated rFGF4 had reduced capacity to induce dimerization of FGFR1c ectopically expressed on the surface of L6 myoblasts (Supplementary Fig. 10i).

We further corroborated the diminished mitogenic character of rFGF4 in the context of wild type mice by treating these mice daily for 15 days. In parallel experiments, we administered mFGF4 and rFGF19 as positive controls; in both cases, these proteins are known to induce full proliferative activity[2,4,37]. Consistent with published literature[38,39], rFGF19 treatment led to dramatic increase in the expression of hepatic proliferation markers (Supplementary Fig. 10j, k). Relative to buffer control, mFGF4 also caused significant increase in the expression of hepatic proliferation biomarkers albeit to significantly lesser degree than rFGF19. Consistent with the data in *db/db* mice, the intensities of proliferation markers in the rFGF4 group were not statistically different from the PBS control group (Supplementary Fig. 10j, k). Notably, the in vivo half-life of rFGF4 (~4.4 h) remained similar to that of mFGF4 (~4.3 h) (Supplementary Table 1) ruling out the possibility that a difference in mitogenic potentials between these molecules was due to reduced biostability of rFGF4. We conclude that rFGF4 is a potent and potentially clinically useful glucose-lowering agent, and that it causes neither hypoglycemia nor other overt adverse effects under our experimental conditions.

To further assess the ability of chronic rFGF4 administration to exert glycemic control, *db/db* and DIO mice were subjected to an oral glucose tolerance test (OGTT). In both cases, we found a sustained reduction in serum glucose level over the course of 2 h (Fig. 5e, f). These results were corroborated via a hyperinsulinemic/euglycemic clamp test in which *db/db* and DIO mice were treated for 14 days with rFGF4. Under hyperinsulinemic conditions, *db/db* mice required more exogenous glucose to maintain euglycemia, as shown by the glucose infusion rate (GIR) (Fig. 5g, h). Moreover, in

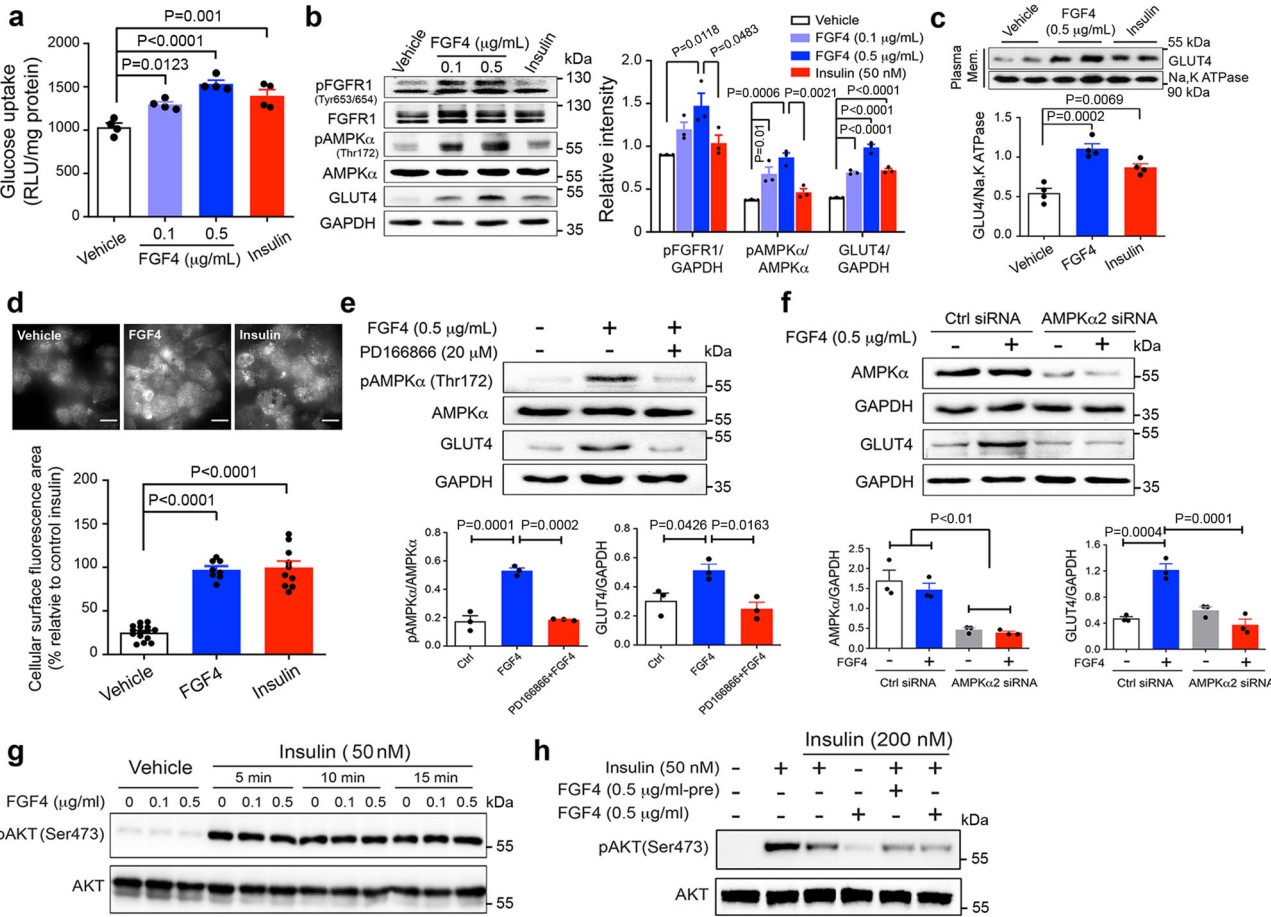

**Fig. 4 rFGF4 enhances GLUT4 expression in L6 myoblasts via activation of the FGFR1/AMPKα signaling pathway. a** Glucose uptake in differentiated rat myoblasts cells (L6) stimulated with rFGF4 (0.1 or 0.5 μg/mL) or a buffer control (vehicle) for 1 h with insulin (100 nM) served as a positive control. Data from four independent measurements are presented as mean +/− SEM. **b** Semi-quantitative western blot analyses of phosphorylation levels of FGFR1 and AMPKα together with expression levels of GLUT4 in L6 cells exposed to rFGF4 (0.1 or 0.5 μg/mL), insulin (50 nM) or a buffer control for 20 min. Data from three independent measurements are presented as mean +/− SEM. **c** Semi-quantitative western blot analyses of plasma membrane GLUT4 protein levels in L6 cells exposed to rFGF4 (0.5 μg/mL), insulin (50 nM) or a buffer control for 20 min. Na, K ATPase was used as a plasma membrane fraction marker. Data from four independent measurements are presented as mean +/− SEM. **d** Plasma membrane translocation of GLUT4 in 3T3-L1 adipocytes transfected with Myc-GLUT4-GFP expression plasmid and stimulated with rFGF4 (0.5 μg/mL) (n = 8), insulin (100 nM) (n = 10) or a buffer control (vehicle) (n = 15) for 20 min shown by total internal reflection fluorescence (TIRF) images. The fluorescent area from GLUT4 dots on the plasma membrane was quantified as mean pixel intensity per unit area and represented as % relative to insulin treatment. Scale bars, 20 μm. Data are presented as mean +/− SEM. **e, f** Semi-quantitative western blot analyses of phosphorylation levels of AMPKα together with expression levels of GLUT4 in L6 cells either pretreated with PD166866 (20 μM) for 30 min and then exposed to rFGF4 (0.5 μg/mL) for an additional 20 min (**e**) or transfected with control or AMPKα2 siRNAs and exposed to rFGF4 (0.5 μg/mL) for 20 min (**f**). Data from three independent measurements are presented as mean +/− SEM. **g** Phosphorylation levels of AKT in L6 myoblasts pretreated with rFGF4 (0, 0.1, or 0.5 μg/mL) for 20 min and stimulated with insulin (50 nM) for another 15 min. The images are representative of three independent experiments with similar results. **h** L6 myoblasts were pretreated with either insulin alone (200 nM), insulin plus rFGF4 (0.5 μg/mL -pre) or a buffer control for 2 h, and then stimulated with either insulin (50 nM), rFGF4 (0.5 μg/mL), or insulin plus rFGF4 for 20 min. The images are representative of three independent experiments with similar results. Phosphorylation levels of AKT in these cells were measured by western blotting. Statistical comparisons in (**a–f**) are one-way ANOVA tests with Tukey's multiple comparisons tests. Source data are provided as a Source data file.

similarly treated DIO mice, there were evident increases in GIR, glucose disposal rate (GDR) and insulin-stimulated GDR (IS-GDR) (Fig. 5i-k) without any significant change in insulin level or suppression of free fatty acids (FFA) (Supplementary Fig. 11). Taken together, these data demonstrate that chronic treatment with rFGF4 substantially alleviates hyperglycemia and improves insulin resistance in T2D and obese mice.

**Chronic rFGF4 treatment promotes skeletal muscle glucose disposal.** As in the case of acute treatment, chronic treatment with rFGF4 led to sustained phosphorylation of AMPKα and increased expression of GLUT4 in the skeletal muscle of *db/db* or DIO mice (Fig. 6a, e). Consistent with improved insulin

resistance, AKT activity in skeletal muscle was also enhanced in these experiments (Fig. 6a, e). This contrasts with the data observed in mice receiving single-dose treatment, in which the insulin signaling was not activated. Activation of the AMPKα and insulin signaling pathways led to increased skeletal muscle glucose disposal and diminished phosphorylation of glycogen synthase (GS) (Fig. 6a, e), an effect that was mirrored by enhanced glycogen storage (Fig. 6b, c and f). Muscle glucose metabolic activity was also enhanced as revealed by an increase in transcription of two key glycolytic enzymes (i.e., *Pkm* and *HK2*), a tricarboxylic acid cycle enzyme (*Sdha*), an enzyme involved in oxidative phosphorylation (*Pdha1*, a subunit of pyruvate dehydrogenase (PDH)), and a decrease in the transcription of *Pdk4*

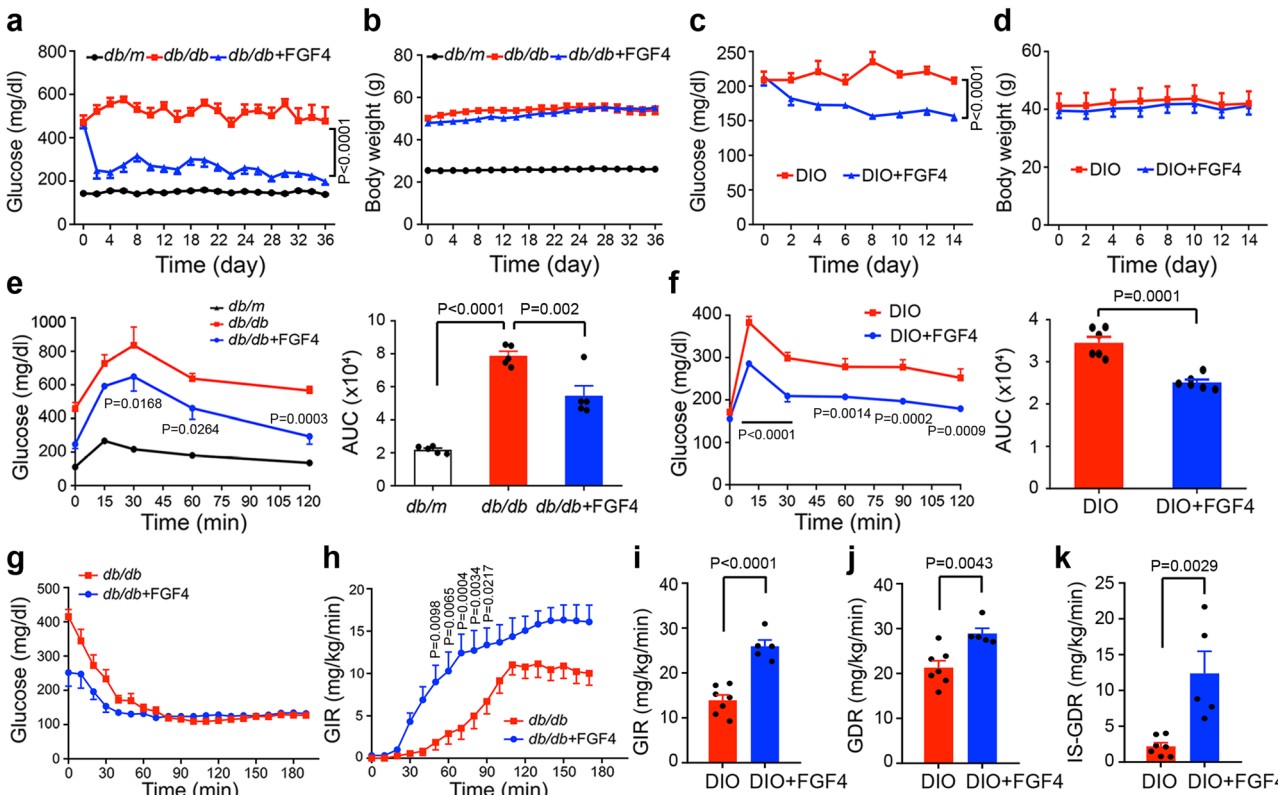

**Fig. 5 Long-term rFGF4 treatment improves glucose intolerance and insulin resistance in diabetic mice. a**, **b** Blood glucose (**a**) and body weight (**b**) of ad libitum-fed *db/db* mice injected with rFGF4 (1.0 mg/kg body weight) (*n* = 8) on alternate days for 37 days with PBS buffer (*db/db*) (*n* = 8); *db/m* mice served as controls (*n* = 14). **c**, **d** Blood glucose (**c**) and body weight (**d**) of ad libitum-fed DIO mice injected with rFGF4 (1.0 mg/kg body weight) on alternate days for 15 days with PBS buffer (DIO) as controls (*n* = 5). **e** Left: OGTT over the course of 120 min done following 37 days of chronic administration of rFGF4 (1.0 mg/kg body weight) to *db/db* mice (*n* = 5). Right: integrated AUC for changes in blood glucose levels. **f** Left: OGTT over the course of 120 min done following 15 days of chronic administration of rFGF4 (1.0 mg/kg body weight) to DIO mice (*n* = 6). Right: integrated AUC for changes in blood glucose levels. **g**, **h** Blood glucose levels (**g**) and glucose infusion rate (GIR) (**h**) during a hyperinsulinemic euglycemic clamp for 14 days in rFGF4-treated (1.0 mg/kg body weight) *db/db* mice (*n* = 6) with PBS buffer (*db/db*) served as controls (*n* = 7). **i**–**k** Glucose infusion rate (GIR) (**i**), glucose disposal rate (GDR) (**j**), and insulin stimulated glucose disposal rate (IS-GDR) (**k**) during a hyperinsulinemic euglycemic clamp for 14 days in rFGF4-treated (1.0 mg/kg body weight) DIO mice (*n* = 5) with PBS buffer (DIO) served as controls (*n* = 7). Data are presented as mean +/− SEM. Statistical comparisons in (**a**, **c**, **e** (left penal) and f (left penal)) are two-way ANOVA tests with Tukey's multiple comparisons tests (**a** and **e**) or Šídák's multiple comparisons tests (**c**, **f** and **h**). Statistical comparison in (**e** (right penal) is one-way ANOVA test with Tukey's multiple comparisons test. Statistical comparisons in (**f** (right penal) and **i**–**k**) are unpaired two-tailed tests. Source data are provided as a Source data file.

(a negative regulator of PDH) (Fig. 6d, g)[40]. We conclude that long-term administration of rFGF4 maintains glucose homeostasis through sustained activation of AMPKα and amelioration of insulin resistance in skeletal muscle, whereas in the case of short-term treatment, rFGF4 maintains glucose homeostasis solely via AMPKα activation.

**rFGF4 ameliorates inflammation by direct action on macrophages.** Adipose tissue macrophage (ATM) infiltration and inflammation play primary roles in the etiology of insulin resistance[41,42]. In obesity, F4/80+CD11c+ macrophages account for the majority of infiltrating ATMs[41,43,44]. We therefore assessed the inflammatory status of Epi-WAT in chronically rFGF4-treated *db/db* mice by immunostaining with antibodies to CD68 (Cluster of Differentiation 68, a protein highly expressed in macrophages) and F4/80. We noted reduced expression of CD68 and monocyte chemotactic protein 1 (MCP-1) in Epi-WAT excised from rFGF4-treated animals (Fig. 7a–c). Consistent with these observations, we found a substantial decrease in F4/80+ macrophages in rFGF4-treated *db/db* mice relative to controls (Fig. 7d). Immunofluorescence using F4/80 and CD11c-specific antisera showed a marked increase in F4/80+CD11c+

macrophages in *db/db* mice, a feature that was greatly diminished by treatment with rFGF4 (Fig. 7d, e). There was also a corresponding decrease in transcription of several pro-inflammatory genes in Epi-WAT, including *KC, Galectin-3, IL-1β, IL-6, CCl2, CCl3, CCl5, Cxcl10*, and *TNF-α* (Fig. 7f).

To corroborate these findings, we did an in vivo macrophage chemotaxis assay which entailed injecting GFP-labeled monocytes into *db/db* mice treated with either buffer alone or rFGF4 (Fig. 7g). In this experiment, rFGF4 administration resulted in a substantial decrease in the number of GFP-labeled adipose tissue monocytes (Fig. 7h). Moreover, ATM infiltration and accumulation in Epi-WAT was greatly reduced by chronic rFGF4 treatment, an observation presumably attributable to a decrease in F4/80+CD11c+ macrophages (Fig. 7i). Consistent with reduced inflammation in Epi-WAT in rFGF4 treated *db/db* mice, we found reduced expression of inflammatory genes in skeletal muscle and in liver (Supplementary Fig. 12a, b). rFGF4 treatment also resulted in reduced serum levels of some pro-inflammatory cytokines (i.e., IL-6 and TNF-α) (Fig. 7j), as well as of alanine aminotransferase (ALT) and aspartate aminotransferase (AST) (Supplementary Fig. 12c) in *db/db* mice. Reminiscent of the *db/db* mice subjected to chronic treatment with rFGF4, there were also decreases in expression of CD68 and

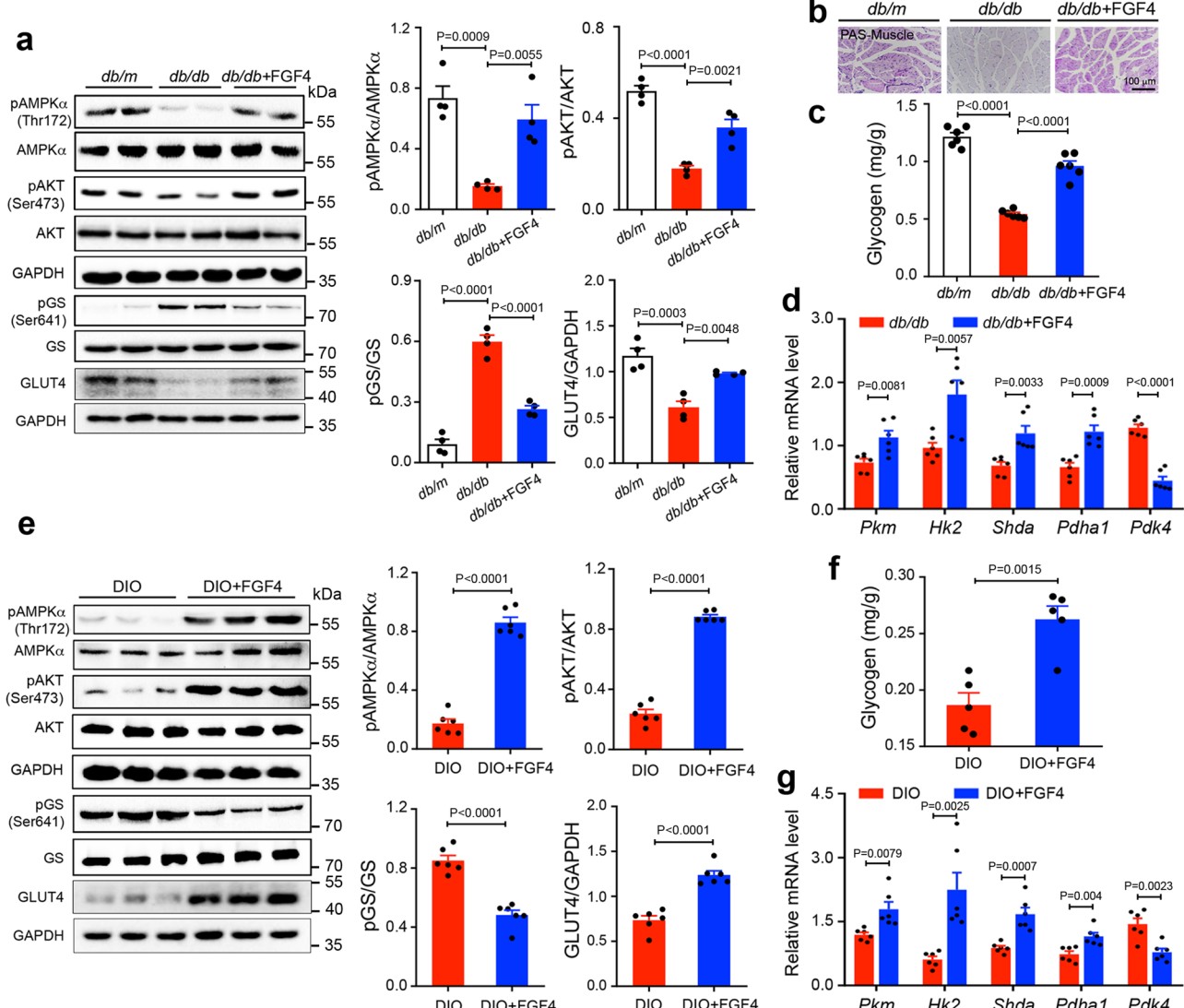

**Fig. 6 Long-term rFGF4 treatment improves skeletal muscle glucose disposal in *db/db* and DIO mice. a–d** *db/db* mice were treated every other day for 37 days with rFGF4 (1.0 mg/kg body weight) or a buffer control; littermate *db/m* mice served as additional controls. **a** Phosphorylation levels of AMPKα, GS, and AKT and expression levels of GLUT4 as determined by western blotting (left panel) and quantitation using ImageJ software (right panel) (*n* = 4). **b**, **c** Representative periodic acid–Schiff staining (PAS; magenta denotes glycogen) (**b**) and glycogen content (**c**) of skeletal muscle in *db/db* mice (*n* = 6). **d** Real-time PCR analysis of expression of *Pkm, Hk2, Shda, Pdha1,* and *Pdk4* mRNAs (*n* = 6). Scale bars, 100 μm. **e–g** DIO mice were treated every other day for 30 days with rFGF4 (1.0 mg/kg body weight) or a buffer control. **e** Phosphorylation levels of AMPKα, GS, and AKT and expression levels of GLUT4 were determined in skeletal muscle by western blotting (left panel) and quantified using ImageJ software (right panel) (*n* = 6). **f** Glycogen content of skeletal muscle in DIO mice (*n* = 5). **g** Real-time PCR analysis of expression of *Pkm, Hk2, Shda, Pdha1,* and *Pdk4* mRNAs (*n* = 6). Data are presented as mean +/− SEM. Statistical comparisons in (**a**, **c**) are one-way ANOVA tests with Tukey's multiple comparisons tests. Statistical comparisons in (**d–g**) are unpaired two-tailed tests. Source data are provided as a Source data file.

inflammatory genes in Epi-WAT as well as reduced serum levels of IL-6 and TNF-α in long-term rFGF1-treated *db/db* mice (Supplementary Fig. 13a-c).

Given that macrophages endogenously express FGFR1c[45] and that AMPKα activation in macrophages ameliorates adipose tissue inflammation[46], we considered the possibility that rFGF4 might act directly on these cells to attenuate the inflammatory response via activation of the AMPKα pathway. To test this idea, we exposed cultured primary macrophages to high glucose and examined the phosphorylation of FGFR1, AMPKα, IKKα/IKKβ, IκBα, and JNK in response to rFGF4 treatment. We found that rFGF4 enhanced phosphorylation of FGFR1 and AMPKα, while reducing phosphorylation levels of JNK and IKKα/β in a dose-dependent manner (Fig. 7k). This was accompanied by a corresponding suppression in

dissociation of the IκBα/NF-κB complex, and by phosphorylation of IκBα (Fig. 7k). Moreover, the immunofluorescence assay showed that nuclear import of the NF-κB subunit P65 was strongly suppressed by rFGF4 treatment (Fig. 7l), with a reduction of IL-6 in the macrophage culture medium following incubation with rFGF4 (Fig. 7m). These data demonstrate that rFGF4 can alleviate ATM infiltration and inflammation in diabetes by acting directly on macrophages.

## Discussion
FGF4 plays indispensable roles during embryonic development, including implantation, morphogenesis, and organogenesis. In adults, it is expressed at low level in duodenum, ileum, and colon where it is proposed to maintain intestinal stem cells[27]. At the

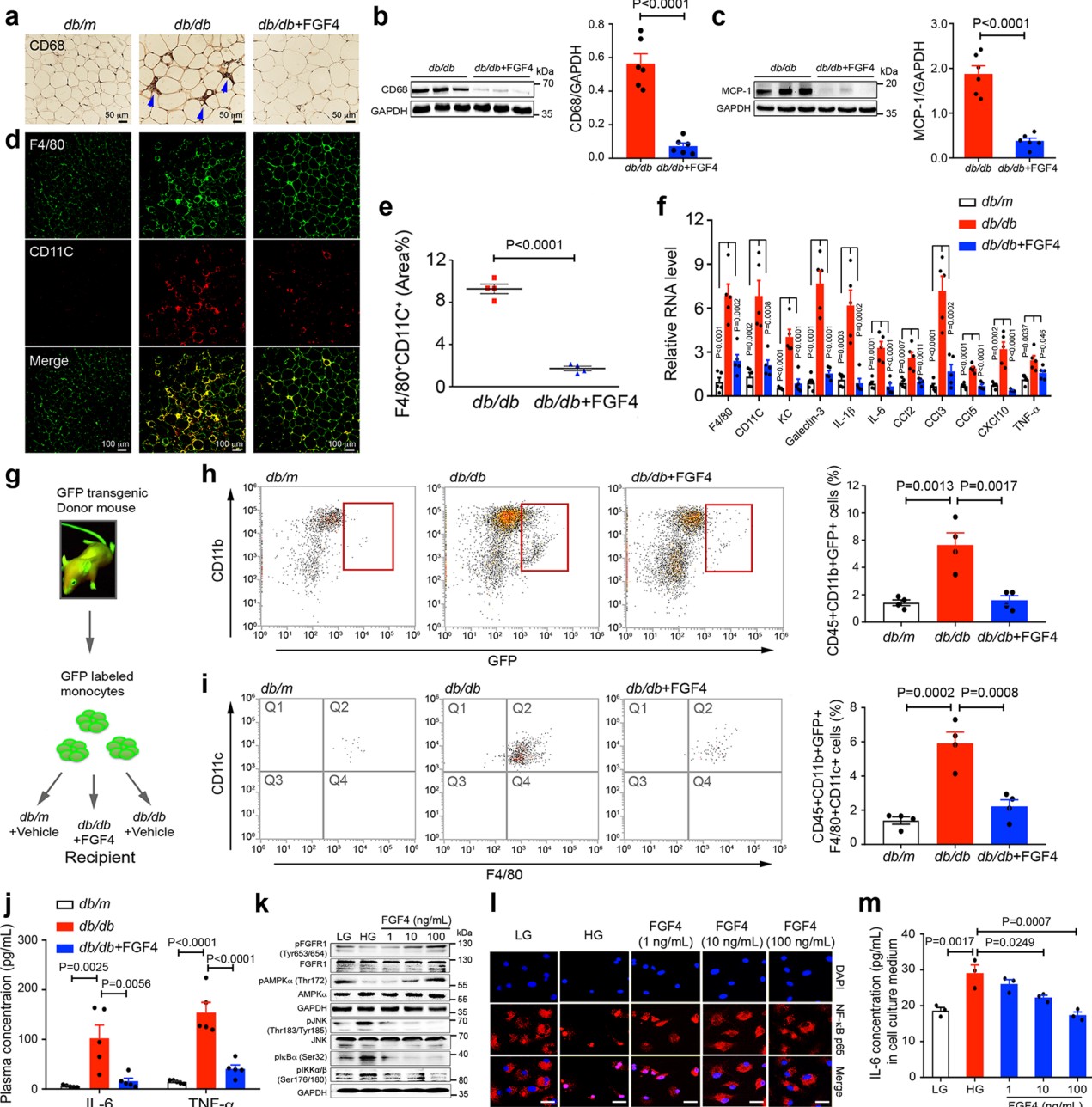

**Fig. 7 Long-term rFGF4 treatment alleviates local and systemic inflammation in *db/db* mice. a–j** Analysis of Epi-WAT tissue and sera from *db/db* mice treated over the course of 37 days with rFGF4 (1.0 mg/kg body weight). **a** Immunohistochemical staining with an anti-CD68 antiserum. Data are representative of 4 mice from each group. Scale bars, 50 μm. **b**, **c** Western blot analyses showing expression of CD68 (**b**) and MCP-1 (**c**) in Epi-WAT tissue and their quantitation (shown at right) (*n* = 6). **d** Immunofluorescence staining with antibodies to either F4/80 or CD11C (n = 4). Scale bars, 100 μm. **e** Quantification of immunofluorescence staining with antibodies to either F4/80 or CD11C in panel d using ImageJ software. **f** Real-time PCR analysis of expression of *F4/80, CD11c, KC, Galectin-3, IL-1β, IL-6, CCl2, CCL3, CCl5, CXCl10*, and *TNF-α* mRNAs (*n* = 5). Littermate *db/m* mice served as controls. **g** Schematic of in vivo macrophage tracking experiments. **h**, **i** FACS analysis of total CD45⁺CD11b⁺GFP⁺ monocytes (**h**) and total CD45⁺CD11b⁺ GFP⁺F4/80⁺CD11c⁺ macrophages (**i**) in mouse stromal vascular cells (*n* = 4). **j** Serum concentrations of TNF-α and IL-6 determined by ELISA (*n* = 5). **k** Western blot analyses of bone marrow-derived macrophages (BMMs) pretreated with rFGF4 (1, 10, or 100 ng/mL for 1 hr) and exposed to either low glucose (LG) or high glucose (HG) (36 mM) for 2 h. The images are representative of three independent experiments with similar results. Antisera used were against phosphorylated or unphosphorylated forms of FGFR1, AMPKα, JNK, IKBα, and IKKα/β; an antibody to GAPDH was used as a loading control. **l** Immunofluorescence staining showing nuclear import of NF-κB subunit P65 (red) in BMMs incubated with LG or HG and treated with increasing concentrations of rFGF4. The images are representative of three independent experiments with similar results. Nuclei were stained with DAPI (blue). Scale bars, 25 μm. **m** Expression of proinflammatory cytokine IL-6 as determined by ELISA in BMMs pretreated with rFGF4 (1, 10, or 100 ng/ml) for 1 h, followed by treatment with HG (36 mM) for an additional 24 h. Data from three independent measurements are presented as mean +/− SEM. Statistical comparisons in (**b**, **c** and **e**) are unpaired two-tailed tests. Statistical comparisons in (**f**, **h**, **i**, **j** and **m**) are one-way ANOVA tests with Tukey's multiple comparisons tests. Source data are provided as a Source data file.

cellular level, it acts by regulating cell proliferation and differentiation of embryonic and tissue stem cells in a paracrine fashion[2,37,47]. It exerts its diverse functions by binding, dimerizing, and hence activating its multiple cognate FGFRs including FGFR1c, FGFR2c, FGFR3c splice isoforms, and FGFR4 with the assistance of HS[2,37,47]. FGF4 shows greatest activity towards FGFR1c followed by FGFR2c, FGFR3c, and FGFR4[19].

In this study, we screened multiple paracrine FGFs in a search for those that possess a glucose-lowering effect in T2D mice. This search resulted in the discovery of FGF4 as a potent anti-hyperglycemic factor, with glucose-lowering effects rivaling those of FGF1 (Fig. 1 and Supplementary Fig. 1). Our findings expand the potential armamentarium of FGFs for the treatment of T2D and related metabolic disorders. Notably, unlike FGF1, which causes a phasic restriction of food intake, FGF4 does not affect food intake (Fig. 1 and Supplementary Fig. 8e). This may be attributed to the fact that FGF1 is the most promiscuous FGFR ligand capable of binding and activating all seven principal FGFRs, including the 'b' splice isoforms[19,48,49]. In contrast, FGF4 does not cross-react with 'b' splice isoforms at all.

We found via micro-PET CT that short-term administration of either rFGF4 or rFGF1 in mice led to conspicuous glucose uptake in skeletal muscle, and to a lesser extent in liver and in WAT (Fig. 2a–c). Because skeletal muscle lacks β-klotho (a mandatory co-receptor of endocrine FGF21[4]), rFGF21 is unable to induce $^{18}$F-FDG uptake in this tissue (Fig. 2a–c). These results are consistent with previous reports that endocrine FGF21 and FGF19 mediate their glucose-lowering activities principally in liver and WAT, but not in muscle[4,10,50]. Therefore, the inability of FGF19 and FGF21 to stimulate glucose uptake in skeletal muscle—the main site of insulin-mediated glucose disposal (up to ~75%)[51,52]—accounts for the inferior potency of these endocrine FGFs relative to paracrine FGF4/FGF1, which can target this β-klotho deficient tissue.

GLUT4 is the key modulator that facilitates glucose uptake in the skeletal muscle[22,23]. Indeed, dysregulation of GLUT4 has been associated with insulin resistance in skeletal muscle. Although GLUT4 level is moderately decreased in skeletal muscle of obese rodents, impaired insulin-dependent GLUT4 translocation in this tissue is considered as one of the main reasons for obesity-induced insulin resistance[53–56]. Targeted overexpression of GLUT4 in the skeletal muscle is considered as a potential approach to improve insulin resistance[23,57]. It has been suggested by a number of studies that recruitment of GLUT4 to the cell surface of skeletal muscle is acutely activated by signals including insulin and AMPK[24,25,29,58,59]. Insulin elicits GLUT4 translocation via PI3-kinase-dependent mechanisms whereas exercise relies on the activation of AMPK signaling[29,59]. Exercise activates AMPK probably through increasing the AMP/ATP ratio or activating LKB1 or Ca$^{2+}$ associated CaMKK2[59,60]. In addition to its role in GLUT4 translocation, activated AMPK is thought to be translocated to the nucleus, where it stimulates binding of MEF2 to the GLUT4 promotor[61] or phosphorylates histone deacetylase 5[62] to upregulate GLUT4 expression. Here we found that acute treatment of db/db mice with rFGF4 triggers AMPKα activation rather than activating the insulin-dependent pathway to subsequently upregulate GLUT4 expression and translocation to the cell membrane (Fig. 3 and Supplementary Fig. 3). Consistent with our observations, activation of AMPKα has recently been shown to induce an acute glucose-lowering effect without enhancing insulin signaling[40,63]. Furthermore, we found that rFGF4 activates AMPKα via activation of CaMKK2 probably as a result of intracellular Ca$^{2+}$ release following engagement of FGFR1 (Supplementary Fig. 7). Indeed, suppression or elimination of skeletal muscle AMPKα2 in insulin-resistant mice abrogated FGF4-mediated GLUT4 translocation and expression as

well as FGF4's glucose-lowering effect (Fig. 3 and Supplementary Fig. 4).

It has been reported that activation of AMPKα in adipose tissue reduces macrophage infiltration and associated inflammation[64]. A particularly striking finding with regard to the anti-diabetic effect of rFGF4 is its direct action on macrophages, where it blocks the inflammatory response and tissue inflammation in liver, muscle, and adipose tissue (Fig. 7 and Supplementary Fig. 12), likely via activation of AMPKα signaling. Because inflammatory cytokines block insulin signaling and induce insulin resistance directly[65], our observed decrease in macrophage infiltration and inflammation explains the enhanced insulin-sensitizing effect of chronic rFGF4 treatment. Given the ability of rFGF4 to induce glucose uptake in skeletal muscle and to ameliorate systemic inflammation, our data offer promise that rFGF4 may yield a clinically useful agent for the treatment of T2D.

## Methods

**Expression and purification of FGFs and FGFR1c ectodomain**. Recombinant full length mature human FGF1 (rFGF1) (Met1-Asp155), full length mature form human FGF4 (mFGF4) (Ala31-Leu206) and its N-terminally truncated human FGF4 (rFGF4) (Ala67-Leu206), full length mature human FGF7 (rFGF7) (Cys32-Thr194), FGF8 (rFGF8) (Gln23-Arg215), FGF9 (rFGF9) (Met1-Ser208), FGF19 (rFGF19) (Leu25-Lys216), FGF21 (rFGF21) (His29-Ser209) and soluble extra-cellular D2–D3 region of human FGFR1c (Asp142-Arg365) were all expressed in E. coli (BL21) and purified as described in Supplementary experimental procedures.

**Surface plasmon resonance spectroscopy**. All real-time biomolecular interactions were performed on a BIAcore T200 system (GE Healthcare, Piscataway, NJ) in HBS-EP buffer (10 mM HEPES-NaOH, pH 7.4, 150 mM NaCl, 3 mM EDTA, 0.005% Surfactant P20) as described in Supplementary experimental procedures.

**Animals and animal welfare**. All animal protocols used in our studies were approved by the Animal Care and Use Committee of Wenzhou Medical University, China. We have complied with all relevant ethical regulations for animal testing and research.

Four- to eight-week-old male db/db (C57BLKS/J-lepr$^{db}$/lepr$^{db}$) mice and their non-diabetic db/m littermates, male ob/ob (C57BL/6J-lep$^{ob}$/lep$^{ob}$) mice, male EGFP transgenic (STOCK-Tg (CAG-EGFP)/Nju) mice, male C57BL/6J mice (wild type mice) and male Sprague Dawley (SD) rats were purchased from the Model Animal Research Center of Nanjing University (Nanjing, China). Male and female FGFR1$^{flox/flox}$ (B6.129S4-Fgfr1$^{tm5.1Sor}$/J) mice were purchased from The Jackson Laboratory. Male and female AMPKα2 knockout mice[66] were obtained from Dr. Louise D. McCullough at the Department of Neurology, McGovern Medical School, University of Texas Health Science Center at Houston (UTHealth), Houston, TX, USA. Four- to six-week-old male CaMKK2 knockout mice were obtained from Dr. Shengcai Lin at the School of Life Sciences, Xiamen University, Xiamen, Fujian, China. To confirm genotypes, genomic DNA prepared from tail snips was analyzed using PCR. All protocols used in our studies were approved by the Animal Care and Use Committee of Wenzhou Medical University, China. All animals were acclimatized to our laboratory environment before use and housed in a specific pathogen-free (SPF) animal facility with controlled environment (22 ± 2 °C, 50–60% humidity, 12-h light/dark cycle, lights on at 7 AM) and free access to food and water. Diet-induced obese (DIO) mice were established by feeding male C57BL/6J mice, AMPKα2 knockout mice, or CaMKK2 knockout mice with a high-fat diet (20% protein; 60% fat; 20% carbohydrates; Cat. D12492; Research Diets, Inc, New Brunswick, NJ) and with free access to water from an age of 4–6 weeks. After 12 weeks feeding on either a standard or high-fat diet, mice were randomly assigned by body weight and fed blood glucose levels determined using a FreeStyle complete blood glucose monitor (Abbott Diabetes Care Inc., Alameda, CA).

Four- to six-week-old male FGFR1$^{flox/flox}$ mice were fed with a high-fat diet for 8 weeks and randomly assigned by body weight and fed blood glucose levels. Conditional skeletal muscle FGFR1 knockout mice were established by local gastrocnemius muscle injection of HBAAV2/9-CMV-cre-GFP viruses with HBAAV2/9-GFP viruses as controls (purchased from Hanbio Biotechnology Co., Ltd., Shanghai, China). These mice received ~2 × 10$^{10}$ viral particles per injection and eight injections per limb, and were continuously fed with a high-fat diet for 4 weeks.

**Glucose-lowering studies of rFGF1, rFGF4, rFGF7, rFGF8, and rFGF9 in db/db mice**. For analysis of the acute (single dose) effects of rFGF1, rFGF4, rFGF7, rFGF8, and rFGF9 on glucose levels in db/db mice, 30 animals were placed into 5 groups of 6 mice each and each group injected intraperitoneally (i.p.) with rFGF1,

rFGF4, rFGF7, rFGF8, or rFGF9 (1.0 mg/kg body weight). Blood samples were taken from tail veins 6 h after treatment and blood glucose was measured as described above.

**Functional evaluation of rFGF1, rFGF4, and rFGF21 in diabetic mice**. Before each study, mice were randomized based on body weight and blood glucose level. All drugs and buffer controls were delivered by i.p. injection into mice unless specified otherwise. Full details of animal protocols (i.e., acute and chronic efficacy evaluation) are described in Supplementary experimental procedures.

**Indirect calorimetry**. *db/db* mice were treated with rFGF4 (1.0 mg/kg body weight) on alternate days over a 37-day period with buffer (PBS) as a control. The animals were then acclimated for 24 h in metabolic cages before monitoring. Both light cycle and dark cycle measurements of all animals were recorded at 49-minute intervals for 4 days. Oxygen consumption ($VO_2$), carbon dioxide production ($VCO_2$), respiratory exchange ratio (RER), heat production, food intake, and total and ambulatory activity were measured by an indirect calorimetry system (Oxymax, Columbus Instruments, Columbus, OH).

**Hyperinsulinemic-euglycemic clamp study**. *db/db* or DIO mice received either rFGF4 (1.0 mg/kg body weight) or buffer alone as a control on alternate days over 2 weeks. Hyperinsulinemic-euglycemic clamp tests were performed as previously described[42] with minimal modification. Briefly, one catheter (Silastic 508-001, Dow Corning) was implanted in the right jugular vein of anesthetized mice, then tunneled subcutaneously to the back of the neck and exteriorized. Mice were allowed to recover for 4 days prior to clamp experiments. Mice losing <5% of their pre-surgery weight were included in the test, described in full in Supplementary experimental procedures.

**RNA extraction, cDNA synthesis, and quantitative RT-PCR**. Total RNA was extracted from mouse tissues with TRIzol reagent (Thermo Fisher Scientific, Waltham, MA). Details of cDNA synthesis and qRT-PCR are described in Supplementary experimental procedures.

**Micro positron emission computed tomography (microPET-CT) imaging and analysis**. Overnight fasted *db/db* mice were injected i.p. with rFGF1, rFGF4, rFGF21 (each at 1.0 mg/kg body weight), or PBS (as a control). After 2 h, mice received a single dose (177.6 +/− 20.65 μCi, i.v.) of $^{18}$F-FDG through the lateral tail vein and were anaesthetized (via inhalation of 1.5% isoflurane in oxygen at 2 L/min) and scanned during a 1-h period in an Inveon preclinical multimodal PET-CT (Siemens, Munich, Germany). Images were reconstructed using a three-dimensional ordered subsets expectation maximum (3D OSEM) algorithm followed by Maximum a Posteriori (MAP). 3D regions of interest (ROIs) were drawn over the liver and muscle guided by CT images; radiological activity of corresponding tissues was measured using an Inveon Research Workplace (Siemens, Munich, Germany). After scanning, mice were sacrificed by cervical dislocation. The liver, lower limb muscles (thigh muscles), back muscle (erector spinae muscles), Epi-WAT, and SWAT were excised and radiotracer uptake (ID%/g) in these tissues was measured by γ-counting. Aliquots of the injected dose were set aside and counted contemporaneously with tissue samples to correct for radioactive decay.

**Pathological, histopathological, immunohistochemical, and immunofluorescent evaluation of mouse tissues**. Epi-WAT, liver or skeletal muscle tissues were excised from wild type, FGFR1$^{flox/flox}$, *db/m*, *db/db*, conditional skeletal muscle FGFR1 knockout, or AMPKα2 knockout mice following acute or chronic administration of indicated proteins (all at 1.0 mg/kg body weight) or with PBS buffer as a control and weighed. Epi-WAT, liver, and skeletal muscle were fixed in 4% paraformaldehyde overnight and embedded in either paraffin or using Tissue-Tek OCT compound (Sakura, Tokyo, Japan). After deparaffinization and rehydration, paraffin sections (5 μm) were stained with haematoxylin and eosin (H&E) reagent using standard procedures. Details of immunohistochemistry and immunofluorescence analysis of paraffin sections are described in Supplementary experimental procedures. Glycogen content of skeletal muscle tissue (100–200 mg) from *db/db* or DIO mice treated for 37 or 30 days with PBS (as a control) or rFGF4 was determined according to the manufacturer's instructions (Abcam, Cambridge, UK).

**Pharmacokinetic evaluation**. The in vivo half-lives of rFGF1, rFGF4, and mFGF4 was determined following a single i.p. injection (all at 1 mg/kg body weight) into adult male SD rats (220–250 g). Detailed pharmacokinetic analyses of rFGF1, rFGF4, and mFGF4 are described in Supplementary experimental procedures.

**Experiments using L6 muscle cells**. L6 rat skeletal myoblasts (Serial number: GNR 4, National Collection of Authenticated Cell Cultures of China, Shanghai, China) were used for myotube formation. Details of cell preparation and stimulation are described in Supplementary experimental procedures.

**Proximity ligation assay**. L6 rat skeletal myoblasts ectopically expressing human FGFR1c were used for proximity ligation assay. Details of cell preparation and stimulation are described in Supplementary experimental procedures.

**Experiments using 3T3-L1 adipocytes**. 3T3-L1 mouse embryonic fibroblasts (Resource number: ATCC® CRL-3242™, ATCC, Manassas, Virginia) were used for adipocyte differentiation. Details of cell preparation and stimulation are described in Supplementary experimental procedures.

**Calcium imaging**. Mouse embryonic fibroblasts (MEFs) were established by introducing SV40 T antigen into primary cultured embryonic cells from mice at E13.5. For determining the rFGF4-induced calcium influx, MEFs grown on 35-mm glass-bottom dishes (70% confluence) were treated with Fluo-4-AM (final concentration, 5 μM) (Molecular probes, Thermo Fisher Scientific, Waltham, MA) for 30 min, washed twice with PBS, and incubated in fresh medium. The fluorescent intensity at time zero was recorded. Cells were then treated with rFGF4 (0.5 μg/ml) or buffer control for 10 min and change in fluorescence intensity was recorded using a confocal microscope (Zeiss, Oberkochen, Germany).

**Cell proliferation assay**. HepG2 cells (obtained from Dr. Shengcai Lin at the School of Life Sciences, Xiamen University, Xiamen, Fujian, China) (Resource number: ATCC® HB-8065™, ATCC, Manassas, Virginia) were seeded in 96-well plates at $3 \times 10^3$ cells/well in 100 μl medium and maintained in a 5% $CO_2$ incubator for 24 h at 37 °C. After serum starvation for 12 h, the cells were treated with increasing concentration of mFGF4 or rFGF4 (0–500 ng/ml) for 24 h at 37 °C. The cell proliferation assay was performed using cell counting kit-8 (CCK-8, Beyotime Biotechnology, Shanghai, China); 10 μl CCK-8 reagent was added to each well and cells were incubated for 1 h at 37 °C. The absorbance at 450 nm was measured using a microplate reader (Molecular Devices, CA, USA).

**In vivo monocyte tracking**. EGFP labeled monocytes were obtained from an EGFP transgenic donor mouse (STOCK-Tg (CAG-EGFP)/Nju) and lysed in red blood cell (RBC) buffer. Monocyte subsets were enriched with an EasySep® mouse monocyte enrichment kit according to the manufacturer's instructions (STEMCELL Tech, Vancouver, BC). Isolated monocytes ($2 \times 10^6$–$5 \times 10^6$) were washed once in serum-free DMEM medium, resuspended in PBS buffer, and their densities determined. $0.5 \times 10^6$ viable cells were suspended into 0.2 ml PBS and injected into *db/m* or *db/db* mice via the tail vein. After treatment with either buffer (PBS) or rFGF4 (1.0 mg/kg body weight) on alternate days for a 10-day period, stromal vascular cells (SVCs) were isolated from Epi-WAT and analyzed by Flow Cytometry.

**Stromal vascular cell (SVC) isolation and FACS analysis**. Epididymal fat pads from *db/m* or *db/db* mice were minced and digested in PBS buffer containing 1 mg/ml type II collagenase (Sigma-Aldrich, St. Louis, MO) and 2% BSA (pH 7.4) at 37 °C for 30 min. The cell suspension was filtered through a 100 μm cell strainer and centrifuged at $1500 \times g$ for 10 min. The floating adipocyte fraction was removed from the SVC pellet. The pellet was incubated with RBC lysis buffer (Thermo Fisher Scientific, Waltham, MA) for 5 min, re-centrifuged at $1500 \times g$ for 10 min, and the pellet resuspended in PBS buffer supplemented with 2% FBS. Finally, SVCs were incubated with FcR blocking reagent (Miltenyi Biotec, Bergisch Gladbach, Germany) at 4 °C for 20 min, followed by an incubation with fluorochrome-conjugated antibodies to CD45 BV510, CD11b PE, CD11c BV421 (Biolegend, San Diego, CA), or F4/80 APC (Thermo Fisher Scientific, Waltham, MA). Labeled cells were analyzed using FACSAria II (BD Biosciences, NewJersey, USA) and data analysis was done using Flow Jo (Tree Star) (Supplementary Fig. 14).

**Bone marrow-derived macrophage (BMM) based experiments**. The isolation and culture of primary BMM from C57BL/6J mice was performed as previously described;[67] experimental details are provided in Supplementary experimental procedures.

**Western blot analysis**. L6 myoblasts, bone marrow-derived macrophages, or mouse tissues were homogenized in RIPA lysis buffer (25 mM Tris, pH 7.6, 150 mM NaCl, 1% NP-40, 1% sodium deoxycholate, 0.1% SDS) supplemented with protease and phosphatase inhibitors (all from Thermo Fisher Scientific Waltham, MA). Plasma membrane proteins of L6 muscle cells were isolated according to the manufacturer's instructions provided (Invent Biotechnologies, Inc, Minnesota, USA). Protein concentrations were determined using a BCA Kit (Protein Assay Kit, Beyotime Biotechnology, Shanghai, China). Equal quantities of soluble protein (40 μg) were separated using 8–12% SDS-PAGE and electro-transferred onto a nitrocellulose membrane. Protein blots were probed with antibodies against primary antibodies (refer to *Reagent and Resource-Antibody*, Supplementary Table 3). Immune-reactive bands were detected by incubating with secondary antibody (Santa Cruz Biotechnology, Dallas, TX; or Biosharp, Hefei, China) conjugated with horseradish peroxidase and visualized using enhanced chemiluminescence (ECL) reagents (Bio-Rad, Hercules, CA). The optical densities of the immunoblots were

analyzed using ImageJ image software (version 1.38e, NIH, Bethesda, MD) and normalized to the scanning signals of their respective controls.

**Statistical analysis**. In vitro experiments were repeated at least in triplicate. The results were expressed as mean ± SEM. Statistical analysis was performed using GraphPad Prism 7 built-in tests. For experiments with two groups, Student's unpaired $t$ tests were used. In the comparison of mean values of more than two groups, one-way or two-way ANOVA with post-tests (Tukey or Šídák) were used. $P < 0.05$ was considered to be statistically significant.

**Reporting summary**. Further information on research design is available in the Nature Research Reporting Summary linked to this article.

## Data availability

All data supporting the key findings of this study are included within the main article and its supplementary materials and can be obtained from the corresponding authors upon reasonable request. Source data are provided with this paper.

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

## Acknowledgements

This work was supported by Grants from National Key R&D Program of China (2017YFA0506000, 2017YFA0205400) (to X.L., Z.H., and P.L.), Natural Science Foundation of China (92057122, 81874323, 81770800, 81622010 to Z.H. and P.L.), CAMS Innovation Fund for Medical Sciences (2019-12M-5-028, 2016-I2M-4-001, 2021-1-I2M-016 to X.L. and P.L.), Beijing Outstanding Young Scientist Program (BJJWZYJH01201910023028 to P.L.), Key Project from Science Technology Department of Wenzhou (88920006, 2021ZY0064 to Z.H.) and the College Students' innovation of Science and Technology Activities Plan of Zhejiang Province (2019R413086) (to Y.H.).

## Author contributions

L.Y., L.W., K.G., Y.H., N.L., S.W., X.L., Q.Z., J.Z., L.Z., J.N., C.C., L.S., S.H., L.K., J.R., P.L., and Z.H. researched the data. X.L., J.R., P.L., M.M., and Z.H. contributed to the initial discussion and design of the project. L.Y., M.M., and Z.H. wrote the manuscript.

## Competing interests

The authors declare no competing interests.
