## [Peer Review File · Nature Communications]

Reviewers' Comments:

Reviewer #1:

Remarks to the Author:

For the most part, I believe the authors do a good job of responding to reviewer's comments pretty thoroughly. However, I still feel that the novelty/significance of this work is marginal without evidence for the physiological role of FGF4 on metabolism. Otherwise, this is just another pharmacological mimetic for FGF1. The mechanistic insights into the role of skeletal muscle in mediating the glucose lowering effects of FGF4, which is contrasted and different to the glucose lowering effects of FGF21, is interesting. Finally, I don't particularly care for the significant adjustments in the brightness and contrast to most of the blots in the paper.

Minor comment:

- 1) In Figure Figure 1C-E, I believe the y-axis units are incorrect (should be mg/dL).
- 2) Several citations are lacking including "as previously reported for FGF21" (line 112).

Reviewer #4:

Remarks to the Author:

This revised manuscript describes the effects of pharmacological treatment of genetically and diet-induced obese mice with FGF isoforms. While it is clear that FGF4 and FGF1 are able to improve glycemic control in these mouse models. A mechanism is proposed that FGF4 increases AMPK signaling. This, in turn, increases both GLUT4 protein expression and GLUT4 translocation. There are some important questions that must be answered before this data set can be properly interpreted. The authors might consider alternative mechanisms to explain the improved glycemic control with pharmacologic FGF4 treatment

Major questions:

1. The effects of FGF4 expression are surprising. After only 6-hours of expression there is a surprising large increase in protein expression, perhaps 8-fold, although it is not a quantitative measurement. This indicates a rather profound increase in protein translation in the phase of a profound activation of AMPK activity. It is confusing how protein translation would be so profoundly increased in the face of AMPK activation, a pathway that minimizes energy-requiring anabolic pathways. Later, with chronic FGF4 infusion, the increase in GLUT4 protein is around 2-fold. This is more reasonable increase for muscle GLUT4 expression as a larger increase would likely lead to hypoglycemia.
2. It is unclear how total GLUT4 levels are increased, but the GLUT4 is not responding to insulin (referring to the statement in the abstract that insulin does not recruit GLUT4 to the cell surface in the FGF4-treated cells).
3. The Myc-GLUT4-GFP staining in 3T3-L1 cells did not look like a typical reporter assay carried out in this cell type. Generally, the cell surface staining is more evenly dispersed on the cell surface and does not have a profoundly punctate appearance.
4. The muscles that were studied in mice were not properly described. Lower limb muscles and back muscles are too vague. Which muscle groups were studied?
5. The authors have misrepresented the existing literature with regard to the downregulation of

GLUT4 muscles in db/db mice and diet-induced obese mice. Depending on the muscle group, there is only a small reduction in total GLUT4 protein. Insulin-dependent GLUT4 translocation, but not exercise-induced translocation, is reduced in obese mice.

6. It is unclear by FGF4 does not affect glycemic control in lean mice. What is the basis for this interaction between obesity and FGF4 action?

7. The western blots need to have the molecular weights added. Also, were the protein extracts used for immunolabeled with the GLUT4 antibody boiled prior to loading on the SDS gel? Heat treating the GLUT4 samples causes the GLUT4 protein to aggregate and not enter the gel. Many commercial antibodies label a non-specific band that remains after a sample is boiled.

Reviewer #5:

Remarks to the Author:

The authors described paracrine FGF4 as a novel anti-hyperglycemic FGF. Using diabetic mice model, they found that both paracrine FGF4 and FGF1 are more effective in lowering blood glucose level than endocrine FGF21. The reason for this is that FGF4 and FGF1 target skeletal muscles, which express FGFR1c but lack β -Klotho, an obligatory FGF21 co-receptor. They also showed that FGF4 treatment upregulates GLUT4 abundance on the cell surface in skeletal muscle. FGF4 activates AMPK α after binding to FGFR1c, in a manner that is independent of insulin signaling. The authors also demonstrated that FGF4 acts directly on macrophages by blocking inflammation response in multiple metabolic tissues. Unlike FGF1, FGF4 had no effect on food intake. FGF4 can therefore be considered a promising agent for use in the treatment of type 2 diabetes and other metabolic disorders.

Within the revision process the authors have improved the manuscript significantly. They have changed the Introduction and Discussion section according to the suggestions of Reviewer 2 and 3, added additional controls required by Reviewer 2 and specified some experimental details as suggested by Reviewer 3. Most of the Reviewers' comments have been taken into account. However, some points have not been fully clarified (please see in bold).

Regarding the comments of Reviewer 3:

General comment

Although the manuscript covers an interesting topic, it is rather challenging to understand how the experiments were performed and how conclusions were drawn e.g. inferior potency of FGF21 in lowering blood glucose when no direct comparison between rFGF4 and FGF21 were performed. In the introduction, FGF4 is very succinctly mentioned, it will be helpful to describe what is known about FGF4, for example: where it is expressed? how it signals? Receptor requirements (which FGFR)? Also, this manuscript will benefit by clarifying the method section (e.g. justify doses for the compounds utilized, timing of the compounds administration). The authors decided to combine results and discussion into one section, however the data presented in the manuscript is not being discussed/compared to previously published data. Lastly, FGF4 is a member of the fibroblast growth factor family, a group of genes which has a broad range of biological involvements: embryonic development, cell growth, tissue repair, tumor growth and invasion. It would be important for understanding of the value of a FGF4-based therapy via lowering blood glucose to demonstrate if this approach will be safe.

In the revised manuscript, the Authors have compared the anti-hyperglycemic effects of FGF4, FGF1 and endocrine FGF21, revealing more potent glucose lowering effect for FGF1 and FGF4 than for FGF21 (Fig 1.). They have also extended the Discussion section and added more detailed description of the experimental methods in the Supplementary Information.

Regarding the reviewer's comment on safety, the Authors claim that their recombinant truncated FGF4 protein has been shown to be non-mitogenic in vivo. They have showed in the livers of wild type mice that after 15 days of treatment with FGF4 the intensity of one proliferation marker, Ki67, was not statistically different from the PBS control group (Supplementary Fig. 9f, g). ****The second proliferative marker PCNA seems not to be different between PBS-, full-length FGF4- and

truncated FGF4-treated groups. This issue should be clarified. Moreover, the authors could have presented data from in vitro studies using classical mitogenic experiments. My main concern is the lack of comparison of full-length and truncated FGF4 variants, as well as a deeper explanation of the mechanism of action of truncated FGF4. If it is still able to activate FGFR1c, why it is not mitogenic? Is it due to a change in ligand-receptor interaction, a different dimerization mechanism, or simply the kinetics or stability of the FGFR-FGF4 complex?

Specific questions/comments:

1) Does FGF4 binds to FGFR1c only? Please determine if it interacts with the other FGF receptors?
Figure 1: Please reorder the figure logically, as FGF8 should be shown in panel 1d before FGF9.

The Authors have fully responded to the Reviewer's comment and provided specific references. Using conditional knockout of FGFR1 in skeletal muscle, they found that the ability of FGF4 to regulate glucose is mediated through its binding to FGFR1.

2) Figure 1i: serum of FGF4 were significantly reduced in DIO-L not in DIO

The Authors have reorganized the manuscript and removed the pathophysiological data, focusing primarily on the pharmacological and mechanistic study of FGF4 to alleviate hyperglycemia in T2D. In my opinion it is the better option.

3) In the experiment reported in Figure 2 did you measure food intake? If so, please report. Did you measure insulin and lipid levels? Also did you measure body weight and organ weights?

The Authors have fully responded to the Reviewer's comment and clarified the experiment presented in Figure 2. They have also performed additional experiments, showing the effects of FGF4 on body weight and organ weights (Supplementary Table 2), and analyzing serum insulin levels in wild type mice (Supplementary Fig. 1b and 1d).

4) What is the half-life of rFGF4?

The Authors have measured the half-life of FGF4 at 1 mg/kg of body weight in SD rats and found that it is ~4.4 hours (Supplementary Table 1). Unfortunately, they have not compared the obtained value to the parameters of full-length FGF4 and FGF1.

5) Figure 2: Can you justify how you determined which time point to select to assess efficacy: 2, 3 or 6 hours after rFGF4 administration? Why were 2 independent 18F-FDG experiments conducted (Figure 2f and 2g)?

The Authors' response is well justified. The selected time points were chosen based on time-series analysis of the glucose-lowering effect. The difference in the experimental setting presented in Fig 2f and 2g have been clearly explained.

6) Supplementary Fig3a-c: FGF4 and FGF21 were not directly compared in the same experiment, therefore specify more clearly that they were not tested side by side. FGF21 has been shown to rapidly decrease blood glucose in ob/ob and DIO mice but not in db/db mice. It would be helpful to determine if FGF4 and FGF21 show differences on glucose lowering in DIO mice.

In the revised version of the manuscript, the Authors have compared anti-hyperglycemic effects of paracrine FGF4 and FGF1 and endocrine FGF21 in parallel over the time course of 36 h treatment (Fig. 1). They found that both FGF1 and FGF4 elicited significant glucose lowering effects 3 hours after administration and normalized the glucose level by 6 hours in db/db, DIO, and ob/ob mice.

7) In the experiment reported in Figure 2j how did you determine which dose of Phloretin to use?

The Authors found in the literature that the dose of phloretin used to antagonize GLUT function ranged from 100 to 500 mg/kg of body weight. In the study, they used 200 mg/kg and found that this concentration of phloretin was sufficient to inhibit most glucose uptake as compared with solvent controls.

8) Same question for the experiment reported in Figure S5 with PD166866?

Due to the fact that PD166866 inhibitor can block the activity not only of FGFR1 but also other FGFR isoforms, the Authors have removed the data showing the effect of inhibitor and substituted them with results obtained in mice specifically lacking FGFR1 expression in skeletal muscle. I agree that these new results more directly verify the key role of FGFR1 in mediating the anti-hyperglycemic effect of FGF4, however in my opinion both types of experiments could have been shown.

9) Supplementary Fig8a and b are important figures. It will be critical to explain how you designed your N-terminal truncation? It is not obvious from this study if rFGF4 full length would have induced liver hyperplasia as it was not run in parallel in the experiment with the N-terminal truncated FGF4 molecule. Also, FGF19 could have been used as a positive control for this experiment. It is important to discuss why PCNA and Ki67 are increased in db/db mice with buffer control when compared to db/m mice?

The Authors used an engineered N-terminal truncation (lacking the first 36 residues) of FGF4 to eliminate its mitogenic activity (P Bellosta et al., Mol Cell Biol, 2001, 21(17):5946-57).

However, according to the cited paper the mitogenic activity of the truncated FGF4 on NIH 3T3 cells was only slightly lower than that of the mature FGF4. This discrepancy has not been mentioned and discussed in the manuscript.

The Authors showed that long-term FGF4 treatment of db/db mice did not generate any signs of hepatic hyperplasia, as revealed by intensities of markers including proliferating cell nuclear antigen (PCNA) and Ki67 (Supplementary Fig. 9a, b). They also claim that in wild type mice, 15 days of FGF4 treatment did not result in mitogenic activity compared with PBS controls, as the intensities of proliferation markers were similar in both groups. However, in the case of PCNA there was no statistical difference between full length FGF4 (mitogenic) and truncated FGF4. Thus, the conclusion that the truncated version of FGF4 lacks mitogenic activity is definitely too far-fetched and not well justified. The statement that truncated FGF4 is not mitogenic in comparison to the full length protein should be corrected.

As suggested by the Reviewer FGF19 was used as a positive control.

Other points not raised by the Reviewers:

1. In Fig. 3 to 7 FGF1 as a positive control would be recommended.
2. It is not clear how Western Blot analysis was performed. Was each detection with specific antibodies performed on a separate membrane (from independent SDS-PAGE) or was the membrane reprobed?

REVIEWER COMMENTS

Reviewer #1 (Remarks to the Author):

For the most part, I believe the authors do a good job of responding to reviewer's comments pretty thoroughly. However, I still feel that the novelty/significance of this work is marginal without evidence for the physiological role of FGF4 on metabolism. Otherwise, this is just another pharmacological mimetic for FGF1. The mechanistic insights into the role of skeletal muscle in mediating the glucose lowering effects of FGF4, which is contrasted and different to the glucose lowering effects of FGF21, is interesting. Finally, I don't particularly care for the significant adjustments in the brightness and contrast to most of the blots in the paper.

We thank the reviewer for appreciating the extensive efforts we put in revising the manuscript. However, we regret that the reviewer thought that our work merely describes another FGF1 mimetic and therefore the novelty/significance of our work is marginal. Contrary to the reviewer's assessment, the purpose of this work is not so much to report identification of FGF4 as another mimetic of FGF1. Rather in this study we identified the underlying physiological mechanism for superior pharmacologic activities of paracrine FGFs relative to the endocrine FGF21 in exerting a glycemic control. Notably our discovery of skeletal muscle as the main target of paracrine FGFs in regulating glucose levels is unprecedented and will major implications in drug discovery for human metabolic diseases.

Minor comment:

1) In Figure Figure 1C-E, I believe the y-axis units are incorrect (should be

mg/dL).

Response: This error has been fixed in the revised manuscript (see Fig. 1C-E in the revised manuscript).

2) Several citations are lacking including "as previously reported for FGF21" (line 112).

Response: We are grateful to the Reviewer for bringing this omission to our attention. We have now cited all the necessary publications in the revised manuscript in support of the statement (page 5, line 116, page 9, lines 225-227 and page 14, line 334).

Reviewer #4 (Remarks to the Author):

This revised manuscript describes the effects of pharmacological treatment of genetically and diet-induced obese mice with FGF isoforms. While it is clear that FGF4 and FGF1 are able to improve glycemic control in these mouse models. A mechanism is proposed that FGF4 increases AMPK signaling. This, in turn, increases both GLUT4 protein expression and GLUT4 translocation. There are some important questions that must be answered before this data set can be properly interpreted. The authors might consider alternative mechanisms to explain the improved glycemic control with pharmacologic FGF4 treatment.

Major questions:

1. The effects of FGF4 expression are surprising. After only 6-hours of expression there is a surprising large increase in protein expression, perhaps 8-fold,

although is not a quantitative measurement. This indicates a rather profound increase in protein translation in the phase of a profound activation of AMPK activity. It is confusing how protein translation would be so profoundly increased in the face of AMPK activation, a pathway that minimizes energy-requiring anabolic pathways. Later, with chronic FGF4 infusion, the increase in GLUT4 protein is around 2-fold. This is more reasonable increase for muscle GLUT4 expression as a larger increase would likely lead to hypoglycemia.

Response: The apparent discrepancy between the effects of the acute and chronic treatment on GLUT4 expression is due to a difference in sample preparation. In earlier experiments (i.e., acute treatment), the cell lysate were boiled at 100°C for 10 mins before loading on the SDS gel for further immunolabeling with GLUT4 antibody. This has resulted in appearance of non-specific bands contributing to the artificial massive increase in GLUT4 expression. Upon consultation with technical staff at Proteintech, Rosemont, Illinois and discussions with our collaborators who provided us with GLUT4 expression data in cultured cells and in AMPK knockout mice (Fig. 3d, Fig. 4 and Supplementary Fig. 5c), we learned that to avoid interference from these non-specific bands samples should not be boiled rather should be gently warmed up to 38°C prior to loading on the SDS gel. Thus, we have redone the western blotting experiments using the suggested sample preparation protocol and indeed our new data show comparable inductions in GLUT4 expression in response to acute and chronic FGF4 treatments (revised Fig. 2g, 3f, 6a and 6e).

2. It is unclear how total GLUT4 levels are increased, but the GLUT4 is not responding to insulin (referring to the statement in the abstract that insulin does not recruit GLUT4 to the cell surface in the FGF4-treated cells).

Response: It has previously been demonstrated that cell surface abundance of GLUT4 in the skeletal muscle is acutely upregulated by two independent signals namely insulin and AMPK (D Grahame Hardie, et al., *Nat Rev Mol Cell Biol*, 2012, 13(4):251-62; Shaohui Huang, et al., *Cell Metab*, 2007, 5(4):237-52; Robert T Watson, et al., *Trends Biochem Sci*, 2006, 31(4):215-22). In our study, we found that rFGF4 alone is capable of upregulating GLUT4 cell surface abundance and subsequently enhancing glucose uptake in skeletal muscle. These rFGF4-mediated effects were severely compromised by AMPK α deficiency. Therefore, we conclude that rFGF4-induced upregulation of GLUT4 cell surface abundance in skeletal muscle is AMPK α -dependent but insulin-independent.

3. The Myc-GLUT4-GFP staining in 3T3-L1 cells did not look like a typical reporter assay carried out in this cell type. Generally, the cell surface staining is more evenly dispersed on the cell surface and does not have a profoundly punctate appearance.

Response: Based on the comment made by the Reviewer, it appears that we have not adequately described our imaging strategy to visualize GLUT4 expression on/near the plasma membrane in adipocytes derived from 3T3-L1 cells. Unlike most previous studies which employed confocal microscopy, we used Total Internal Reflection Fluorescence (TIRF) microscopy, a method that is capable of collecting fluorescent signals within 400 nm from the cell membrane. Notably, in our TIRF imaging experiments we selected a 59.6° angle for excitation laser to ensure that the excited fluorescent signal originated strictly from the narrow zone near cellular surface. Under these experimental settings, we observed punctate expression of GLUT4.

Notably our imaging data are consistent with previously published ones by Frolov and coworkers who analyzed cell surface expression of GLUT4 by

imaging a adipose cell line transfected with GLUT4-GFP (Fig. R1C) by using confocal microscopy and TIRF (Fig. R1D) (cited by Fig. R1 in J Cell Biol. 2005 May 9;169(3):481-9). In this study, confocal images with 3D reconstruction showed an evenly dispersed GLUT4 expression, while TIRF images showed profoundly punctate GLUT4 distribution.

Fig. R1 in J Cell Biol. 2005 May 9;169(3):481-9.

4. The muscles that were studied in mice were not properly described. Lower limb muscles and back muscles are too vague. Which muscle groups were studied?

Response: Thigh muscles of the lower limb and erector spinae muscles of back were analyzed in our study. We have stated these muscle groups in the revised manuscript (Page 6, lines 140-141 and Page 23, lines 567-568).

5. The authors have misrepresented the existing literature with regard to the downregulation of GLUT4 muscles in db/db mice and diet-induced obese mice. Depending on the muscle group, there is only a small reduction in total GLUT4 protein. Insulin-dependent GLUT4 translocation, but not exercise-induced

translocation, is reduced in obese mice.

Response: We thank the Reviewer for making us aware of these facts. Accordingly, we have rewritten the relevant part in the revised Discussion section (pages 17-18, lines 424-427) as 'Although GLUT4 level is moderately decreased in skeletal muscle of obese rodents, impaired insulin-dependent GLUT4 translocation in this tissue is considered as one of the main reasons for obesity-induced insulin resistance'.

6. It is unclear by FGF4 does not affect glycemic control in lean mice. What is the basis for this interaction between obesity and FGF4 action?

Response: Hypoglycemia is a severe side effect of insulin replacement therapy (American Diabetes Association, *Diabetes Care*, 2019, 42(Suppl 1): S90-S102). It has been previously reported that treatment with FGF1 or FGF21, both cognate ligands of FGFR1c, does not induce hypoglycemia in insulin resistant mice (Jae Myoung Suh, et al., *Nature*, 2014, 513(7518):436-9; Alexei Kharitononkov, et al., *J Clin Invest*, 2005, 115(6):1627-35). A similar observation has been made regarding the acute glucose-lowering effect of AMPK activators (MK-8722 and PF-739) in skeletal muscle (Robert W Myers, et al., *Science*, 2017, 357(6350):507-511; Emily C Cokorinos, et al., *Cell Metab*, 2017, 25(5):1147-1159.e10). FGF1, FGF21 and AMPK activators all induce decreases of insulin levels in insulin-resistant mice compared to vehicle controls. In our study, we also found that acute rFGF4 treatment rescued hyperinsulinemia in *db/db* mice (Supplementary Fig. 1b) and had no significant effect on insulin levels in chow-fed wild type mice (Supplementary Fig. 1d). Therefore, it is plausible that FGF4 treatment does not cause hypoglycemia in diabetic mice because it lowers insulin levels.

This line of research is clearly worth further experimentation, as

important details are not yet understood. However, as suggested by the editor, such experiments are beyond the scope of this study.

7. The western blots need to have the molecular weights added. Also, were the protein extracts used for immunolabeled with the GLUT4 antibody boiled prior to loading on the SDS gel? Heat treating the GLUT4 samples causes the GLUT4 protein to aggregate and not enter the gel. Many commercial antibodies label a non-specific band that remains after a sample is boiled.

Response: As requested by the Reviewer, we have added molecular weights in all western blots in the revised manuscript. With regard to sample preparation for GLUT4 western blotting please see our response to comment #1.

Reviewer #5 (Remarks to the Author):

The authors described paracrine FGF4 as a novel anti-hyperglycemic FGF. Using diabetic mice model, they found that both paracrine FGF4 and FGF1 are more effective in lowering blood glucose level than endocrine FGF21. The reason for this is that FGF4 and FGF1 target skeletal muscles, which express FGFR1c but lack β -Klotho, an obligatory FGF21 co-receptor. They also showed that FGF4 treatment upregulates GLUT4 abundance on the cell surface in skeletal muscle. FGF4 activates AMPK α after binding to FGFR1c, in a manner that is independent of insulin signaling. The authors also demonstrated that FGF4 acts directly on macrophages by blocking inflammation response in multiple metabolic tissues. Unlike FGF1, FGF4 had no effect on food intake. FGF4 can therefore be considered a promising agent for use in the treatment of type 2 diabetes and other metabolic disorders.

Within the revision process the authors have improved the manuscript significantly. They have changed the Introduction and Discussion section according to the suggestions of Reviewer 2 and 3, added additional controls required by Reviewer 2 and specified some experimental details as suggested by Reviewer 3. Most of the Reviewers' comments have been taken into account. However, some points have not been fully clarified (**please see in bold**).

We thank the reviewer for valuing the extensive efforts we put in revising the manuscript and for providing additional suggestions for improving the manuscript.

Regarding the comments of Reviewer 3:

Regarding the reviewer's comment on safety, the Authors claim that their recombinant truncated FGF4 protein has been shown to be non-mitogenic in vivo. They have showed in the livers of wild type mice that after 15 days of treatment with FGF4 the intensity of one proliferation marker, Ki67, was not statistically different from the PBS control group (Supplementary Fig. 9f, g). **The second proliferative marker PCNA seems not to be different between PBS-, full-length FGF4- and truncated FGF4-treated groups. This issue should be clarified. Moreover, the authors could have presented data from in vitro studies using classical mitogenic experiments.**

*Response: In response to this valid concern, we have reanalyzed the differences of PCNA immunohistochemical staining in PBS-, full-length FGF4 (mFGF4)- and truncated FGF4-treated groups. We indeed found a statistical difference between PBS- and full-length FGF4--treated groups (p value= 0.0008) or between truncated FGF4- and full-length FGF4--treated groups (p value= 0.0041) (**Fig.R2, inside panel**) but not between PBS- and truncated FGF4--treated groups (p value= 0.1736) when comparing them separately using unpaired t -tests. However, these p values*

became greater than 0.05 when we compared all groups including FGF19 group using One-Way ANOVA analysis with Tukey's multiple comparisons test; Adjusted *p* value between PBS- and full-length FGF4--treated groups is 0.4714 and between truncated FGF4- and full-length FGF4--treated groups is 0.6116, respectively. We speculate that the reason may be due to great values and great degree of dispersion in FGF19-treated groups that was used as a positive control (FGF19 is known to be a potent mitogen for liver tissue *in vivo*)

Fig. R2 Statistical analysis of PCNA positive cells (%) in the livers of wild type mice after daily administration for 15 days with truncated FGF4 (FGF4) or full-length FGF4 (mFGF4).

Per the Reviewer's suggestion, we have compared the mitogenic potencies of truncated and full-length FGF4 on a hepatocyte cell line (HepG2) *in vitro*. As shown in the revised **Supplementary Fig. 10f**, relative to full-length FGF4, truncated FGF4 sustained a dramatic loss in mitogenic activity.

My main concern is the lack of comparison of full-length and truncated FGF4 variants, as well as a deeper explanation of the mechanism of action of truncated FGF4. If it is still able to activate FGFR1c, why it is not mitogenic? Is it due to a change in ligand-receptor interaction, a different dimerization mechanism, or simply the kinetics or stability of the FGFR-FGF4 complex?

Response: The N-terminally truncated non-mitogenic variant of FGF4 was engineered based on our "threshold model" which posits that the nature of FGF

response is primarily controlled by the thermodynamics of FGF-induced cell surface FGFR dimer (Allen Zinkle, et al., *F1000Res*, 2018, 7:F1000 Faculty Rev-872). Specifically, we showed that a strong/stable FGFR dimerization and correspondingly persistent intracellular signaling is required for a mitogenic FGF signal whereas a weak/short-lived FGFR dimer and correspondingly transitory intracellular signal is sufficient to elicit a metabolic FGF response. Based on our model, non-mitogenic yet fully metabolic variants of FGFs can be developed by simply dampening their ability to dimerize their cognate FGFRs. Indeed, we previously successfully developed two fully metabolic but non-mitogenic versions of paracrine FGF1 – one carrying an N-terminal truncation and another carrying mutations in its HS binding receptors (Zhifeng Huang, et al., *Cell Reports*. 2017, 20(7):1717-1728). More recently, we adopted a similar strategy to develop fully metabolic variants of the endocrine FGF19 – one carrying mutations in its HS binding site and another bearing mutations in its KLB co-receptor binding site that are devoid of unwanted mitogenic effects of parent wild type molecule (Jianlou Niu, et al., *Proc Natl Acad Sci U S A*. 2020, 117(46):29025-29034).

Akin to the N-terminally truncated FGF1, the N-terminal truncation deletes several key residues necessary for stable FGFR binding and dimerization. Indeed, surface plasmon resonance (SPR) spectroscopy experiments showed that compared to full-length wild type parent molecule, the binding affinity of N-terminally truncated FGF4 variant to FGFR1c isoform is weaker (**Supplementary Fig. 10g, h**). Accordingly, proximity ligation assay (PLA) showed that relative its wild-type parent molecule, truncated FGF4 had reduced capacity to induce FGFR1c dimerization on the surface of L6 myoblasts (**Supplementary Fig. 10i**). All these newly generated data are consistent with the results of *in vivo* and *in vitro* mitogenic experiments (**Supplementary Fig. 10 a, b, f, j and k**), which showed that the mitogenic activity of N-terminally truncated FGF4 was largely compromised (**pages 11-12, lines 270-298**). Therefore, we conclude that the reduced mitogenic activity of truncated FGF4 is attributable to its reduced capacity to dimerize FGFR.

The Authors have measured the half-life of FGF4 at 1 mg/kg of body weight in SD rats and found that it is ~4.4 hours (Supplementary Table 1). Unfortunately, they have not compared the obtained value to the parameters of full-length FGF4 and FGF1.

Response: As requested by the Reviewer, we have measured half-lives of full-length FGF4 and FGF1 (all at 1 mg/kg of body weight) in parallel in SD rats. We found that the half-lives of full-length FGF4 and FGF1 are ~4.3 hrs and ~2.4 hrs, respectively (Supplementary Table S1 in the revised manuscript).

Due to the fact that PD166866 inhibitor can block the activity not only of FGFR1 but also other FGFR isoforms, the Authors have removed the data showing the effect of inhibitor and substituted them with results obtained in mice specifically lacking FGFR1 expression in skeletal muscle. I agree that these new results more directly verify the key role of FGFR1 in mediating the anti-hyperglycemic effect of FGF4, however in my opinion both types of experiments could have been shown.

Response: Per the Reviewer's suggestion, we have reintroduced the data generated using PD166866 inhibitor in the revised manuscript (Supplementary Fig. 5d).

The Authors used an engineered N-terminal truncation (lacking the first 36 residues) of FGF4 to eliminate its mitogenic activity (P Bellosta et al., Mol Cell Biol, 2001, 21(17):5946-57). However, according to the cited paper the

mitogenic activity of the truncated FGF4 on NIH 3T3 cells was only slightly lower than that of the mature FGF4. This discrepancy has not been mentioned and discussed in the manuscript.

The Authors showed that long-term FGF4 treatment of db/db mice did not generate any signs of hepatic hyperplasia, as revealed by intensities of markers including proliferating cell nuclear antigen (PCNA) and Ki67 (Supplementary Fig. 9a, b). They also claim that in wild type mice, 15 days of FGF4 treatment did not result in mitogenic activity compared with PBS controls, as the intensities of proliferation markers were similar in both groups. However, in the case of PCNA there was no statistical difference between full length FGF4 (mitogenic) and truncated FGF4. Thus, the conclusion that the truncated version of FGF4 lacks mitogenic activity is definitely too far-fetched and not well justified. The statement that truncated FGF4 is not mitogenic in comparison to the full length protein should be corrected.

Response: *As explained at the beginning our response to the Reviewer, the lack of statistical difference between the full length FGF4 and its truncated variant with regard to PCNA expression was due to high values and large degree of scattering in FGF19-treated group which was used as positive control. Indeed, compared to its full-length parent molecule, the truncated FGF4 exhibits a severely reduced mitogenic activity on a hepatocyte cell line in vitro (see revised **Supplementary Fig. 10f**).*

*We appreciate the reviewer's comment regarding the discrepancy between our current results and earlier published results with regard to extent of differences in mitogenic potentials of wild type and truncated FGF4 molecules. To address this issue, we measured the mitogenic activities of wild type and truncated FGF4 on HepG2 cells using CCK-8 cell proliferation assay. Consistent with our current in vitro and in vivo results, we observed a major impairment in mitogenic activity of the truncated FGF4 relative to wild type counterpart (see revised **Supplementary***

Fig. 10f). *The reason for earlier observation for only a modest difference between mitogenic activities of wild type and truncated FGF4 molecules is unclear. We speculate this may be due to differences in choice of cell type (HepG2 cell Versus NIH 3T3 cell), assay (CCK-8 Versus thymidine incorporation) and ligand concentrations used (0-500 Versus 0-1000 ng/ml).*

Other points not raised by the Reviewers:

1. In Fig. 3 to 7 FGF1 as a positive control would be recommended.

*Response: Per the Reviewer's suggestion, we have included rFGF1 as a positive control in our major findings of rFGF4 in Fig. 3 and Fig.7. As in the case of rFGF4, we found that a single injection of rFGF1 into db/db and diet-induced obese (DIO) mice reduces hyperglycemia by upregulating GLUT4 expression/translocation via FGFR1-dependent activation of AMPK α (**Supplementary Fig. 4d-f and Supplementary Fig. 5**). Likewise, the chronic treatment of db/db mice with rFGF1 ameliorated inflammation in Epi-WAT and reduced serum levels of IL-6 and TNF α (**Supplementary Fig. 13**).*

2. It is not clear how Western Blot analysis was performed. Was each detection with specific antibodies performed on a separate membrane (from independent SDS-PAGE) or was the membrane reprobed?

Response: A single piece of membrane blotted from a single SDS-PAGE gel was used to perform western blotting analysis. Using the molecular weight markers as a guide, the membrane was cut into stripes each covering the one of target proteins. The membrane stripes were then probed with appropriate antibodies to detect expression/phosphorylation.

Reviewers' Comments:

Reviewer #4:

Remarks to the Author:

The response to the previous critiques has satisfied my questions

Reviewer #5:

Remarks to the Author:

In the revision process, the authors significantly improved the manuscript as most of the reviewers' comments were taken into account. However, some issues are still not fully clarified. At this stage, I am not asking for additional experiments to be performed, but only for the text of the manuscript to be improved.

1) The mitogenic potential of FGF proteins regulating glucose homeostasis is a very important issue in terms of their potential pharmacological applications. I therefore asked for clarification of the mitogenic aspect of rFGF4 and the discrepancy between the results presented here and those previously published by Bellosta et al (Mol Cell Biol, 2001, 21(17):5946-57).

To address this issue, Authors measured the mitogenic activity of wild-type and truncated FGF4 on HepG2 cells using the CCK-8 cell proliferation assay (revised supplementary Figure 10f). These results are contrary to previously published observation. I see no reason why the authors chose a different cell model. It is known that NIH3T3 is the most sensitive cell line to FGF-dependent proliferation. To verify this important point, the experimental settings should be the same as previously published.

Consequently, I still believe that the conclusion about the lack of mitogenic activity of the shortened version of FGF4 is far too far-fetched. The statement that truncated FGF4 is not mitogenic compared to the full-length protein should be tempered.

2) The authors claim to have included rFGF1 as a positive control in their main findings for rFGF4 in Fig.3 and Fig.7. I do not see these data in the revised version of the manuscript. Please correct this oversight.

REVIEWERS' COMMENTS

Reviewer #4 (Remarks to the Author):

The response to the previous critiques has satisfied my questions

We thank the reviewer for endorsing the publication of our work.

Reviewer #5 (Remarks to the Author):

In the revision process, the authors significantly improved the manuscript as most of the reviewers' comments were taken into account. However, some issues are still not fully clarified. At this stage, I am not asking for additional experiments to be performed, but only for the text of the manuscript to be improved.

1) The mitogenic potential of FGF proteins regulating glucose homeostasis is a very important issue in terms of their potential pharmacological applications. I therefore asked for clarification of the mitogenic aspect of rFGF4 and the discrepancy between the results presented here and those previously published by Bellosta et al (Mol Cell Biol, 2001, 21(17):5946-57).

To address this issue, Authors measured the mitogenic activity of wild-type and truncated FGF4 on HepG2 cells using the CCK-8 cell proliferation assay (revised supplementary Figure 10f). These results are contrary to previously published observation. I see no reason why the authors chose a different cell model. It is known that NIH3T3 is the most sensitive cell line to FGF-dependent proliferation. To verify this important point, the experimental settings should be the same as previously published.

Response: In the current study, we assessed the mitogenic activities of the mature FGF4 (mFGF4) and its N-terminally truncated variant (rFGF4) *in vivo* exclusively focusing on the liver tissue. We found that mFGF4-treatment of mice led to significant induction of hepatic proliferation markers. In contrast, there was no statistically significant change in intensities of these markers in rFGF4-treated group relative to the PBS control group. To corroborate these *in vivo* findings on liver, we chose the immortalized human hepatocarcinoma cell line, HepG2 to assess the mitogenic activity of rFGF4 *in vitro*. In agreement with the *in vivo* data, rFGF4 was obviously retarded in its ability to induce HepG2 cell proliferation compared to mFGF4. We agree with the Reviewer that in order to develop rFGF4 as a viable therapy for type 2 diabetes and related metabolic diseases, we will have to comprehensively characterize the mitogenic activity of rFGF4 *in vivo* by including multiple mouse organs and utilizing a panel of diverse cell lines *in vitro* including NIH3T3 fibroblasts and other FGF-responsive cells.

Consequently, I still believe that the conclusion about the lack of mitogenic activity of the shortened version of FGF4 is far too far-fetched. The statement that truncated FGF4 is not mitogenic compared to the full-length protein should be tempered.

Response: Per Reviewer's suggestion, we have toned down our statements regarding the mitogenic activity of rFGF4 in the revised manuscript. Specifically, we have replaced "non-mitogenic character" with "diminished mitogenic character" and "lack of the mitogenic activity" with "decrease in the mitogenic activity" (page 11, lines 271 and 274 and page 12, line 287).

2) The authors claim to have included rFGF1 as a positive control in their main findings for rFGF4 in Fig.3 and Fig.7. I do not see these data in the revised version of the manuscript. Please correct this oversight.

Response: Indeed, we did use rFGF1 as a positive control for our main findings on rFGF4 in Figures 3 and 7. The data on rFGF1 were presented in the Supplementary Figures 4d-f, 5 and 13 in the revised manuscript which were apparently missed by the reviewer. For reviewer's convenience, we have attached these supplementary figures below our response. The data show that reminiscent of rFGF4, a single injection of rFGF1 into *db/db* and diet-induced obese (DIO) mice reduces hyperglycemia by upregulating GLUT4 expression and cell surface abundance via FGFR1c-dependent activation of AMPK (Supplementary Fig. 4d-f and supplementary Fig. 5). Moreover, as in the case of rFGF4, the chronic treatment of *db/db* mice with rFGF1 ameliorated inflammation in Epi-WAT and caused a reduction in serum levels of IL-6 and TNF (supplementary Fig. 13).

Supplementary Fig. 4. Both rFGF4 and rFGF1 lower blood glucose levels via activation of AMPK α signaling

(a-f) *db/db* mice were pretreated with either Compound C (an AMPK α inhibitor) at 20 mg/kg body weight or a buffer control 1 hr before receiving an i.p. injection of rFGF4 (a-c) or rFGF1 (d-f) (both at 1.0 mg/kg body weight). (a, d) AMPK α -mediated upregulation of GLUT4 expression in skeletal muscle 6 hrs after rFGF4 (a) or rFGF1 (d) injection as

measured by western blot analysis (left hand panel) and quantitated using ImageJ software (right panels) (n=4). **(b, e)** AMPK α -mediated upregulation of GLUT4 expression and translocation in skeletal muscle 6 hrs after rFGF4 (b) or rFGF1 (e) injection as measured by immunofluorescence staining with an anti-GLUT4 antibody. Data are representative of 4 mice from each group. Scale bars, 50 μ m. **(c, f)** Blood glucose levels of *db/db* mice 6 hrs after rFGF4 (c) (*db/db*+AMPKi (n=4), *db/db*+FGF4 (n=3), *db/db*+FGF4+AMPKi (n=4)) or rFGF1 (n=4) (f) injection. Data are presented as mean +/- SEM. Statistical comparisons in (a, d) are one-way ANOVA tests with Tukey's multiple comparisons tests. Statistical comparisons in (c, f) are two-way ANOVA tests with Tukey's multiple comparisons tests.

Supplementary Fig. 5. A single bolus of rFGF1 lowers blood glucose via FGFR-dependent activation of AMPK signaling

(a) Blood glucose levels before and 6 hrs after i.p. injection of rFGF1 (1.0 mg/kg body weight) into wild type (WT) and AMPK α 2^{-/-} mice (AMPK α 2^{-/-}) fed 12 weeks long with a

high-fat diet (n=8). **(b)** Expression and translocation of GLUT4 in LLM of AMPK α 2^{-/-} mice 6 hrs after i.p. injection of rFGF1 (1.0 mg/kg body weight) shown by immunofluorescence staining using an anti-GLUT4 antibody. Data are representative of 4 mice from each group. Scale bars, 50 μ m. **(c)** GLUT4 expression in LLM of AMPK α 2 knockout mice 6 hrs after i.p. injection of buffer (control) or rFGF1 (1.0 mg/kg body weight) as measured by western blotting (left panel) and quantitated using ImageJ software (n=7). **(d)** Blood glucose levels of *db/db* mice 6 hrs after a single injection of rFGF4 (1.0 mg/kg body weight). Mice were pretreated with the FGFR1 inhibitor (PD166866) (20 mg/kg body weight) or a buffer control 1 hr before receiving an i.p. injection of rFGF4 (n=4). **(e-g)** *db/db* mice were pretreated with PD166866 at 20 mg/kg body weight or a buffer control 1 hr before receiving an i.p. injection of rFGF1 (1.0 mg/kg body weight). **(e)** Blood glucose levels of *db/db* mice 6 hrs after rFGF1 injection (n=5). **(f)** FGFR1-mediated upregulation of GLUT4 expression and translocation in skeletal muscle 6 hrs after rFGF1 injection as measured by immunofluorescence staining with an anti-GLUT4 antibody. Data are representative of 4 mice from each group. Scale bars, 50 μ m. **(g)** Phosphorylation of AMPK α and expression of GLUT4 in skeletal muscle 6 hrs after rFGF1 injection as measured by western blot analysis and quantitated using ImageJ software (n=4). Data are presented as mean \pm SEM. Statistical comparisons in (a, d and e) are two-way ANOVA tests with Tukey's multiple comparisons tests. Statistical comparisons in (c, g) are one-way ANOVA tests with Tukey's multiple comparisons tests.

Supplementary Fig. 13. Long-term rFGF1 treatment improves local and systemic inflammation in *db/db* mice.

(a-c) Analysis of Epi-WAT tissue and sera from *db/db* mice treated over the course of 37 days with rFGF1 (1.0 mg/kg body weight). **(a)** Immunohistochemical staining with an anti-CD68 antiserum. Blue arrowheads represent cells stained with anti-CD68 antiserum. Data are representative of 4 mice from each group. Scale bar, 50 μ m. **(b)** Real-time PCR analysis of expression of *F4/80*, *IL-1 β* , *IL-6*, *CCL2* and *TNF- α* mRNAs. Littermate *db/m* mice served as controls (n=6). **(c)** Serum concentrations of TNF- α and IL-6 determined by ELISA (n=4). Data are presented as mean \pm SEM. Statistical comparisons in (b, c) are one-way ANOVA tests with Tukey's multiple comparisons tests.